# Structure of the 30S translation initiation complex coupled to paused RNA polymerase and its potential for riboregulation

Johann J. Roske [1], Giulia Paris[1], Akanksha Goyal[2], Marina Rodnina [2], Nikolay Zenkin [3], Katarzyna J. Bandyra [1,4] ✉ & Ben F. Luisi [1] ✉

In many bacterial species, transcription and translation can be coupled physically, with potential impact on the rates and efficiency of gene expression. Here, we present structural evidence from cryo-EM demonstrating that a bacterial RNA polymerase that is paused proximally to the promoter can associate with the pioneering 30S translation initiation complex (30S IC). These findings suggest that the physical link between transcription and translation can be established prior to commitment to protein synthesis. Although the mRNA is embedded in this 'early expressome' complex, it can nonetheless interact with small regulatory RNA (sRNA) and be targeted for cleavage in the protein-coding region by the RNA degradosome assembly in vitro. The potential tagging of transcripts with sRNA during pioneering and subsequent stages of translation initiation, when the 30S IC is at the 5' end of a polyribosome, may in principle contribute to efficient and rapid termination of gene expression in response to regulatory signals.

In prokaryotic cells, the cytoplasmic milieu and chromatin intermingle. Consequently, the machinery of transcription and translation can interact and operate concurrently on the same nascent mRNA molecule[1–4]. Transcription-translation coupling (TTC) appears to occur extensively in representative gram-negative bacterial species, but its extent can be affected by growth conditions and varies in different bacterial lineages. For example, in some firmicutes, there may be comparatively fewer encounters between RNA polymerase (RNAP) with the ribosome and TTC may be infrequent[5–7]. The physical and kinetic basis of TTC has been studied extensively in *Escherichia coli*, and structural data from cryo-electron microscopy (cryo-EM) has provided details of the component interactions in complexes formed between RNAP, the 30S small ribosomal subunit, and the fully assembled 70S ribosome particle[8–13]. Interactions between the elongating RNAP and 70S ribosomes have been visualised in situ in *Mycoplasma pneumoniae* under conditions of antibiotic stress[14], implicating the in vivo relevance of that transient assembly. In vitro structural and

single-molecule studies of the transcription-translation complexes, referred to as the "expressome", reveal different configurations between RNAP and 70S are possible, e.g., in collided (TTC-A) or coupled (TTC-B) modes, that depend on the spacer length of the linking mRNA transcript and the recruitment of transcription factors NusG and NusA that can bridge the interface of the RNAP-70S to form an assembly[9–11,13,15,16]. A recent study systematically increased the mRNA spacer length up to 20 codons and described the possibility for the spacer to loop out between the physically coupled transcription elongation complexes (TEC) and 70S ribosome in states of long-range coupling or loose coupling (TTC-LC)[16].

The occurrence of TTC likely depends on relative rates of transcription and translation, and the frequency of transcription pausing by the RNAP. After transcription initiation, mRNA transcript elongation is highly processive but discontinuous, with a strictly regulated balance between elongation and transcription pausing[17]. The nascent mRNA itself can directly regulate sequence-dependent transcriptional

[1]Department of Biochemistry, Sanger Building, University of Cambridge, Cambridge, UK. [2]Max Planck Institute for Multidisciplinary Sciences, Göttingen, Germany. [3]Centre for Bacterial Cell Biology, Biosciences Institute, Faculty of Medical Sciences, Newcastle University, Newcastle Upon Tyne, UK. [4]Present address: Faculty of Chemistry, Biological and Chemical Research Centre, University of Warsaw, Warsaw, Poland. ✉e-mail: k.bandyra@uw.edu.pl; bfl20@cam.ac.uk

pausing without requirement for any co-factors[18,19]. Transcript elongation of many genes is paused within the first 200 nts, which serves early regulation mechanisms, including the recruitment of NusG to facilitate TTC formation[20–23]. Promotor-proximal pausing could provide the window of opportunity required to complete translation initiation, which occurs on a timescale of 10−30 s[24–26]. However, it remains unclear whether the establishment of TTC occurs only once the pioneering 70S ribosome catches up and collides with the paused RNAP, which may have detrimental effects on transcription fidelity[27]. Recent structural evidence shows that RNAP can recruit the 30S ribosomal subunit through NusG, potentially as an early step of translation initiation[28]. Translation initiation and ribosome catch-up are proposed to involve the TTC-LC mentioned above, with a long, >11 codons, looped-out mRNA spacer between TEC and ribosome, and with accessibility of the mRNA spacer to regulatory factors[16]. Subsequent coupled transcription and translation involves a short-range, tightly coupled transcription-translation complex (TTC-B), with a different orientation of ribosome relative to TEC, with short, 7–11 codons, mRNA spacer and with little or no accessibility of the mRNA spacer to regulatory factors[16].

Here, we have prepared *E. coli* 30S translation initiation complexes (30S ICs), including initiation factors IF1, IF2 and IF3 and initiator formylmethionyl-tRNA (fMet-tRNA^fMet), and solved cryo-EM structures of the 30S IC in two states. We further assembled 30S ICs on reconstituted TEC with a long nascent mRNA that mimics a state of promoter-proximal transcription pausing. Cryo-EM single-particle analysis shows that NusG establishes a physical link between the 30S IC and the paused RNAP, despite the long (> 70 nts) mRNA spacer and at in vitro concentrations far below the proposed dissociation constant of the NusE-NusG interaction[29]. The obtained complex displays a large degree of flexibility between the TEC and the 30S subunit, similar to previous observations in the absence of NusA and the 50S subunit[28]. The physically coupled complex of paused transcription and translation initiation, or "early expressome", may represent a transient precursor to mature expressomes in vivo.

We explore if the early expressome can be the target for RNA-mediated regulation by testing if the 30S IC-recruited transcript can be accessed by small regulatory RNA (sRNA) that represses the expression of the target gene through base-pairing complementarity with the mRNA. We used the sRNA MicC that directs cleavage and turnover of the *ompD* transcript encoding the outer membrane protein OmpD in *Salmonella* Typhimurium. Assisted by the RNA chaperone Hfq, MicC recognizes *ompD* and induces downstream *ompD*-cleavage by the endoribonuclease RNase E[30,31]. Considering that *ompD*-targeting occurs in the coding region (at codons 23–26) and that the ternary complex of MicC and Hfq on *ompD* does not prevent ribosome engagement or translocation[30], the 30S IC and a TEC positioned downstream of the recognition sequence for MicC could potentially be targets for riboregulation. We observe that the *ompD* transcript in the early expressome can be cleaved by the endoribonuclease RNase E within the multi-enzyme RNA degradosome at a defined site that is directed by the MicC sRNA. Although the rate of mRNA degradation is slower when the target transcript lies within the early expressome, the preference of RNase E for the sRNA-guided cleavage site over alternative, unspecific cleavage positions is increased, suggesting that the early expressome may help expose the seed pairing region in the mRNA for recognition by sRNA and Hfq. Our cryo-EM data indicates that the flexible tether between RNAP and 30S ribosomal subunit can allow the early expressome to accommodate transcript recognition and degradation close to the RNAP. Thus, based on evidence presented here and in other studies, we propose that the establishment of TTC may serve as a transient checkpoint for the nascent mRNA intermediate, which once tagged by sRNA, becomes the target for co-transcriptional degradation.

## Results

### Cryo-EM structure of *E. coli* 30S translation initiation complexes on the *ompD* mRNA

We had previously prepared 30S IC on a 187 nts long fragment of the *ompD* transcript encompassing the 5′-UTR from position −69 up to +118 (with +1 being A in the AUG start codon) and showed that this *ompD*−30S IC can be recognised by MicC-Hfq and targeted for deactivating cleavage at position +83 by the RNA degradosome[32]. Here, we again reconstituted 30S IC on *ompD* together with fMet-tRNA^fMet and initiation factors IF1, IF2 and IF3, and determined their structure by cryo-EM single-particle analysis. The image processing resulted in two distinct structures of the *ompD*−30S IC that have IF2 and either IF3 or initiator tRNA and IF2 recruited (Fig. 1, "Methods", Suppl Fig. 1 and Table 1). Both reconstructions contain well-defined density for the 30S ribosomal subunit, as well as IF1 and the mRNA, which are recruited in previously described positions[33]. The mRNA is bound to the cavity that is formed between head and body regions of the 30S subunit, and the Shine-Dalgarno sequence (SD) in the 5′-UTR forms a duplex with the anti-SD of the 16S rRNA. IF3 is bound at the 30S near IF1 and mRNA (Fig. 1A, B) in an orientation corresponding to a previously reported model of the *T. thermophilus* 30S IC in "state 1"[33]. This arrangement also agrees with a recent report of *E. coli* IF3-bound 30S in which the IF3-CTD is positioned at the P-site[34].

Our second structure, which originates from the same specimen but was reconstructed from a separate particle subset, lacks IF3 but contains IF2 and fMet-tRNA^fMet in addition to IF1 and the 24 nts segment of the *ompD* mRNA (Fig. 1C). The start codon of the mRNA forms triplet base pairs with the anticodon of fMet-tRNA^fMet that is bound in the P-site (Fig. 1D), confirming that the 30S IC is assembled at the AUG of the *ompD* mRNA fragment. The arrangement of the IC components on the 30S subunit is consistent with previously reported models of the *T. thermophilus* 30S IC in "state 4"[33]. Additionally, our cryo-EM density resolves the N2 subdomain of the *E. coli* IF2 (residues 295-370) which attach to helix 16 of the 16S rRNA, resembling reported structures of *P. aeruginosa* IF2 at the 70S-IC[35] (Fig. 1E). 3D variability analysis indicates that the head region of the 30S subunit has a lower degree of mobility with respect to the 30S body, compared to the structure that contains IF3 but lacks fMet-tRNA^fMet (Supplementary Movie 1).

### Ribosomal bS1 protein contacts the 5′-UTR at the 30S shoulder

We observe additional cryo-EM density at the 5′ mRNA exit site that allows unambiguous placement of the first two domains of bS1 (residues 5-176) in an orientation that is in agreement with previous observations[28,36] (Fig. 1C right, 1F and Suppl Fig. 2A). Notably, the relative positions of the first two OB domains of bS1 domains differs most from available 70S ribosome and expressome structures[10,11,37,38], but agrees with reports of the 30S-tRNA-RNAP complex[28], suggesting that bS1 can undergo large conformational changes (Suppl. Fig. 2A).

At lower map-thresholds, the density for bS1 in our *ompD*−30S IC structure further extends from the described binding site along the shoulder of the 30S body towards proteins bS6 and uS11 and to the binding site of IF3 (Suppl. Fig. 2B). The predicted model for the bS1 protein in isolation[39] shows an array of six OB fold-like S1-domains. The observed density in our 30S IC can accommodate two additional S1 motifs (Suppl. Fig. 2B, "OB3" and "OB4") in an elongated arrangement and along with potentially bound 5′-UTR of the mRNA[28]. This is consistent with other reports of bS1 interacting with upstream portions of mRNA on the 30S surface[40] and/or binding of A/U-rich segments that precede the Shine-Dalgarno sequence[41].

### The *ompD* region for seed pairing and deactivating cleavage is accessible in the paused transcription elongation complex

The sRNA MicC recognises a segment in the *ompD* coding region (codons 23–26, nucleotide positions +67 to +78), referred to as the

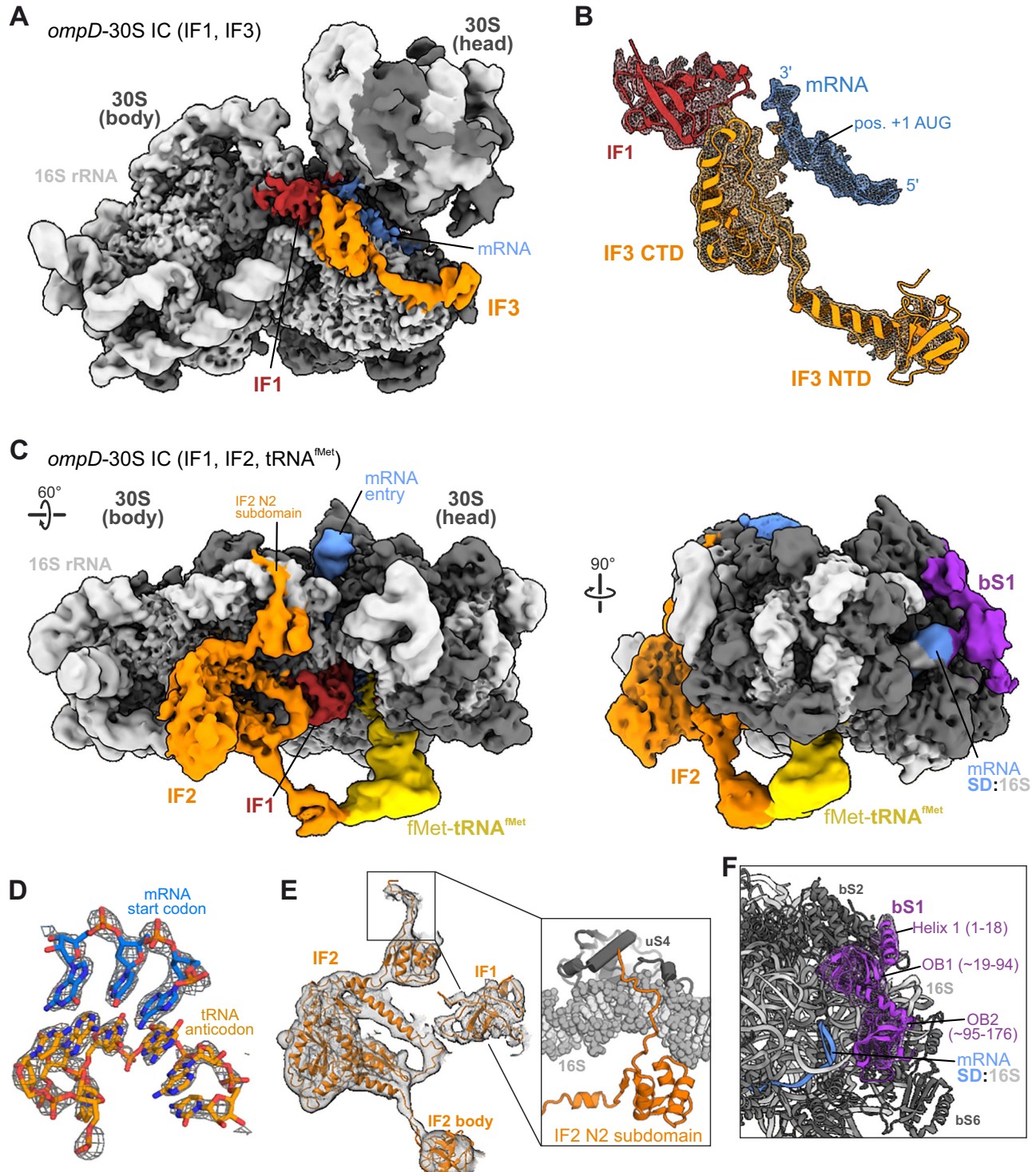

**Fig. 1 | Structure of the *E. coli* translation initiation complex (30S IC) on the *ompD* transcript. A** Cryo-EM map of the 30S IC on the *ompD* mRNA with IF1 and IF3. **B** View of the isolated cryo-EM density around the *ompD* mRNA and IF1/IF3 which are shown as ribbon models. **C** Views of the cryo-EM map of the 30S IC on the *ompD* mRNA with IF1, IF2 and fMet-tRNA^fMet. The rotation symbol on the left indicates the view in respect to (**A**). **D** Close view of the cryo-EM density of the codon-anticodon pair between the *ompD* mRNA and the initiator tRNA. **E** View of the isolated cryo-EM density for IF1 and IF2. Inset: Molecular model for the N2 subdomain of IF2 with the contacting region of the 16S rRNA near ribosomal protein uS4. **F** Molecular model of the first two OB-domains of the ribosomal bS1 protein. The cryo-EM density is shown as a mesh around bS1.

seed pairing region, which leads to *ompD*-cleavage by the endor-ibonuclease RNase E[30,31]. Since MicC and Hfq do not prevent ribosome engagement on *ompD*[30], it is likely that the 30S IC can be formed on *ompD* transcripts that are targeted for degradation. We therefore sought to determine at which stage of gene expression the sRNA can

still exert its regulatory effect. Intriguingly, the ability of RNase E to act on MicC-tagged *ompD* after formation of the 30S IC is dependent on the RNA degradosome assembly, which entails recruitment of DEAD-box RNA helicase RhlB, the glycolytic enzyme enolase and the exori-bonuclease polynucleotide phosphorylase to RNase E[32]. The 30S IC can

exist at the tail of a polyribosome in the context of mature *ompD* transcripts, but also as part of the pioneering ribosome initiating on nascent *ompD* mRNA, upstream of the transcribing RNAP[28].

RNAP can transiently pause during transcription, particularly in promoter-proximal regions, which could help establish the TTC. Genome-wide studies of in vivo transcriptional pausing in *E. coli* have identified a gene-encoded signal element with the consensus motif $G_{-10}Y_{-1}G_{+1}$ (where −1 corresponds to the 3′-end of the nascent RNA, with Y representing a pyrimidine base, Suppl. Fig. 3A) at which transcriptional pausing occurs with increased likelihood[21,42,43]. Adjacent guanine bases at the upstream end (positions $G_{-9}$ and $G_{-11}$ within the motif) could stabilise the DNA-RNA hybrid of a paused TEC in pre- and post-translocated states[21]. Intriguingly, the *ompD* gene harbours this conserved consensus motif with $G_{-10}G_{-9}C_{-1}G_{+1}$ in its coding region, 28 nts downstream of the recognition sequence for the sRNA MicC, which might expose *ompD* to recognition and deactivating cleavage on the paused TEC (Suppl. Fig. 3B). In *S.* Typhimurium *ompD*, the $G_{+1}$ nucleotide of the pause motif lies at position +106 in the coding region and is consistent with a putative paused transcript species of corresponding length that we observed when transcribing *ompD* in vitro using *E. coli* RNAP (Suppl. Fig. 3C).

To investigate how RNAP might impact *ompD* recognition by Hfq/MicC via the seed region at positions +67 to +78 in the *ompD* ORF, and whether a transiently paused TEC could be a potential target for riboregulation via deactivating cleavage by the RNA degradosome at position +83, we reconstituted TECs on different *ompD* segments of increasing length that position RNAP either upstream, at, or downstream of the regulatory region (Fig. 2A and Suppl. Fig. 3D–J). Reconstituted *ompD*-TECs encompass the 5′-UTR (starting from position −69 at the 5′-end) and expose the *ompD* mRNA up to positions +59/+83/+89/+107. The 9 terminal nucleotides at the mRNA 3′-ends were altered to base pair with the same template DNA strand of a short, artificial transcription bubble scaffold for TEC reconstitution (Suppl. Fig. 4A). This approach allowed us to obtain stable complexes suitable for degradation reactions with RNase E as well as cryo-EM studies, which validated *ompD*-TEC assembly on the artificial transcription bubble scaffold (see below).

Cleavage assays of the reconstituted TEC (or the free *ompD* fragment for comparison) with RNase E catalytic domain or full RNA degradosome were undertaken in the presence of MicC and Hfq. RNA fragments were quantified as fractions of respective starting material at time point 0 min. Rates for mRNA degradation and cleavage at position +83 were quantified as described in "Material and Methods". The +72-cleavage product is an off-target species that is only observed in vitro and was disregarded for this analysis[30–32]. We first positioned the *ompD*-TEC upstream of the MicC recognition site, exposing the *ompD* ORF up to position +59, and observed that mRNA and sRNA degradation rates are not affected by the presence of a reconstituted TEC (Suppl. Fig. 3D–F). Next, we placed the TEC so that the *ompD* ORF is exposed up to position +89, mimicking the approximate position of the above-described putative pause element immediately downstream of the MicC hybridisation site. Based on in silico docking with available molecular models of TEC (PDB: 2 ppb) and RNase E (PDB: 2c0b)[44,45], such a reconstituted *ompD*₈₉-TEC is positioned to expose the cleavage site at position +83 with six downstream nucleotides outside the RNAP exit channel and should thus be accessible to the active centre of RNase E. Indeed, we observe the *ompD* intermediate that is produced by MicC-guided cleavage at position +83, as expected (Fig. 2B). In the absence of Hfq-MicC, the +83-cleavage product is not observed and instead the *ompD* fragment is cleaved at off-target position +72. Concurrent to *ompD*-cleavage, MicC is also deactivated by endonucleolytic cleavage at position +9[31]. Thus, the *ompD*₈₉-TEC is accessible to MicC-induced cleavage at position +83 by RNase E as well as the full RNA degradosome complex (Fig. 2D). Moreover, cleavage at position +83 appears more efficient when the *ompD* segment is a part of the

elongation complex, indicated by an increased rate of accumulation of the corresponding intermediate (Fig. 2C and E).

A six nucleotides shorter *ompD*₈₃-TEC should still allow the MicC seed region to hybridize to the recognition sequence in *ompD*, but the +83-cleavage site would be masked by the RNAP exit channel and therefore sterically protected from RNase E cleavage. Indeed, RNase E activity on *ompD*₈₃-TEC at the MicC-induced +83-cleavage site was much less efficient than on free *ompD* in the absence of RNAP (Suppl. Fig. S3G–I), validating correct TEC assembly at the 3′-end of the mRNA and the stability of the complex throughout exposure to regulatory RNA, Hfq and RNase E. However, the deactivation of MicC is still stimulated by the presence of the TEC and the off-target cleavage at *ompD* position +72 is protected by MicC presence, indicating that MicC can hybridise with the seed region in the *ompD*₈₃-TEC. Positioning the TEC further downstream to expose *ompD* up to position +107 does not impact cleavage efficiency at position +83 compared to the same *ompD* fragment in the absence of RNAP (Suppl Fig. S3J), implying that proximity of the RNAP to the regulatory region in *ompD* is a determinant for observed stimulation.

To explore potential direct interactions between RNAP and regulatory factors MicC and Hfq, we determined cryo-EM structures of the *ompD*₈₃-TEC in the presence of Hfq and MicC (Fig. 2F, G and Suppl Fig. 4). The 3D reconstruction shows, at 3.3 Å global resolution, clear density for two α, β and β′ subunits of the RNAP core enzyme, but none for the ω subunit, Hfq or MicC, which may have dissociated from the complex during gel filtration or vitrification. The substrate binding site of the RNAP harbours the duplex DNA construct, which can be continuously traced through the cryo-EM density. The transcription bubble displays the melted base DNA strands from which 10 nucleotides in the template DNA are hybridised to the mRNA and the active site harbours an unpaired template nucleobase with RNAP in the post-translocated state (Fig. 2F). We changed TEC reconstitution and vitrification parameters (see "Material and Methods") and were able to validate the presence of the ω subunit in our RNAP preparation. The resulting cryo-EM reconstructions also display additional density in the vicinity of the mRNA exit channel of the RNAP, likely signal for MicC/Hfq hybridised to the *ompD* recognition site but not adopt in a stable position with respect to the TEC (Fig. 2G). The TEC in these particle classes does not display an altered conformation or additional density on its surface, indicating no direct interaction with the regulatory factors. Nevertheless, the structure shows that the *ompD*-TEC assembles as anticipated and demonstrates that the presence of MicC-Hfq does not disturb the complex.

Our data show that the nascent mRNA becomes accessible to recognition by MicC-Hfq immediately after exiting the RNAP. This is also consistent with the observation that MicC/Hfq protects off-target cleavage of *ompD*₈₃-TEC at position +72, which lies within the seed pairing region and becomes inaccessible to RNase E in the double-stranded mRNA:sRNA hybrid. Furthermore, we show that *ompD* becomes accessible to MicC-induced cleavage by RNase E once the cleavage site exits the TEC. We also observe that a proximally situated TEC increases the efficiency of sRNA-induced RNase E activity and that this effect vanishes when the RNAP is placed further downstream. The putative pause element in *ompD* could also allow the TEC to reside long enough for the pioneering ribosome to initiate translation and possibly establish early TTC. We explore combined effects of paused TEC and initiating 30S ribosome below.

## Cryo-EM structure of a coupled transcription elongation-translation initiation complex

Promoter-proximal pausing of transcription in the coding region of genes after synthesis of the ribosome binding site could allow the translational and transcriptional machineries to synchronise. We explored whether the paused *ompD*-TEC could engage in physical coupling with the pioneering 30S ribosomal subunit during

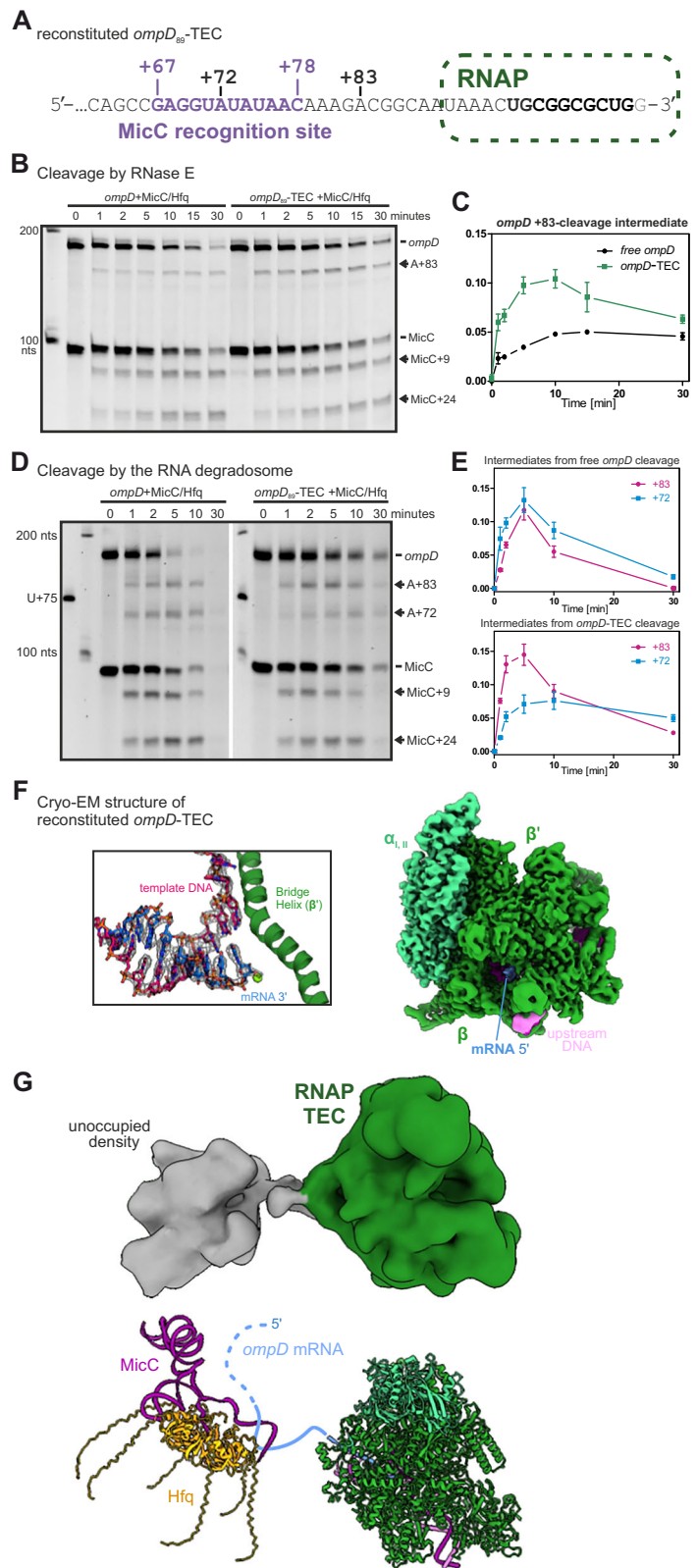

**A** reconstituted *ompD*₈₉-TEC

**B** Cleavage by RNase E

**C** *ompD* +83-cleavage intermediate

**D** Cleavage by the RNA degradosome

**E** Intermediates from free *ompD* cleavage

Intermediates from *ompD*-TEC cleavage

**F** Cryo-EM structure of reconstituted *ompD*-TEC

**G**

recruitment of translation initiation factors. As described above, we have successfully reconstituted TECs and 30S ICs separately and verified that the *ompD*–30S IC assembles at the start codon and demonstrated that the *ompD*-TEC assembles on the artificial transcription bubble at the 3′-end of the *ompD* mRNA fragment, mimicking paused TEC. The TEC has not been assembled on the natural pause-inducing sequence but instead on a scaffold with a modified sequence to

optimise stability for structural studies. We then reconstituted both assemblies together on the *ompD*₈₉ transcript fragment in the presence of NusA, NusG, MicC and Hfq, and performed cryo-EM analysis (Fig. 3A and Suppl Fig. 5). Through iterative rounds of 3D classification, we selected 30S particles that showed clear signal for the initiation factors and initiator tRNA (Suppl Fig. 5). The resulting map can accommodate the 30S IC, mRNA, fMet-tRNA^fMet and IF1-3 in the

**Fig. 2 | The paused *ompD*-transcription elongation complex (TEC) can be directed by small regulatory RNA for cleavage by RNase E. A** Schematic and 3′-sequence of the reconstituted *ompD$_{89}$*-TEC. The 3′-terminal nucleotides that hybridise with the template DNA strand within the TEC are shown in bold black letters. A green oval delineates the 3′-terminal portion of the mRNA that is masked by RNAP in the reconstituted TEC. The recognition sequence for MicC is marked in purple. RNase E cleavage sites observed during in vitro degradation reactions are annotated at positions +72, +83. **B**–**E** Time course reactions of the *ompD$_{89}$*-TEC (or free *ompD* mRNA fragment for comparison) incubated with MicC/Hfq and either RNase E (**B**, **C**) or the full RNA degradosome (**D**, **E**). Cleavage intermediates of *ompD* and MicC are indicated with arrows. The relative abundances of the transient *ompD* cleavage intermediates were quantified (mean ± SD from three independent reactions) and plotted over time in **C** (reactions with RNase E) and **E** (reactions with full RNA degradosome). **F** 3D-refined and locally filtered cryo-EM map of the *ompD*-TEC structure. Subunits of RNAP and bound DNA and mRNA are indicated. The inset shows an isolated view on the RNAP active site. The unwound template DNA strand is shown in red, the nascent mRNA in blue. The bridge helix feature of the RNAP β′-subunit at the active site is depicted in ribbon representation. **G** Structure of *ompD*-TEC with unoccupied density around the mRNA exit site where MicC and Hfq are expected to recognise *ompD*.

positions that we observe in our 30S IC data, and we again observe the elongated density near the mRNA exit site that can be accounted for by the ribosomal bS1 protein (Fig. 3A).

Additionally, our map displays a large area of unoccupied density on top of the 30S head, which is not present in the *ompD*−30S IC structures. To improve the map in this area, we subtracted the signal for the 30S IC from the cryo-EM particles and performed iterative rounds of 3D refinement and classification, which enabled the isolation of a subset of particles that refined to fit the TEC portion of the coupled complex to ~6 Å resolution (Suppl Figs. 5 and 6). The 30S-coupled TEC resembles the above-described structure of the *ompD*-TEC that was solved in the absence of the 30S IC but displays additional density for the ω subunit of RNAP as well as for the N-terminal domain (NTD) of transcription factor NusG. The NusG NTD is bound at the previously described binding site between β and β′ subunits near the unwound non-template DNA strand[46] (Fig. 3A). We also observe well-defined density for the NusG-CTD on the 30S ribosomal subunit near uS10 (NusE)[29], and this density is absent in our structures of the 30S IC that was prepared in the absence of NusG. We observe NusA-NTD contacting the RNAP-portion of the coupled TEC-30S IC complex in a manner similar to previously reported TEC structures containing NusA[47]. A structure of the uncoupled TEC originating from the 30S IC-containing dataset contains additional density for the NusA S1 motif that ensues from the NusA-NTD (Fig. 3A right). We do not observe density for NusA on the surface of the 30S IC, suggesting that this interaction is dispensable for physical coupling between TEC and 30S IC.

Notably, density for the TEC appeared only in 3D-reconstructions generated from the subset of 30S particles that also contained signal for the IFs and fMet-tRNA$^{fMet}$. Additionally, *ompD$_{89}$*-TEC-IC samples prepared in the absence of NusG did not display physical coupling between TEC and 30S IC (Suppl. Fig. 7), suggesting that the physical coupling between RNAP and 30S IC is dependent both on NusG and mRNA-engagement by the small ribosomal subunit.

When the cryo-EM particles of the final subset are aligned on either 30S IC or RNAP, the respective other component remains only visible as noisy density (Fig. 3B and Suppl. Fig. 5). Multi-body Refinement in RELION allows the analysis of the flexibility between TEC and 30S[48]. The first three eigenvectors account for 50.8 % of the variance in the data (19.8, 16.1 and 14.9 % for components 1, 2 and 3, respectively), which is visualised Fig. 3C and Suppl. Movie 2. This high mobility (up to 42° rotation and 87 Å change in distance between mass centres along individual eigenvectors) between TEC and 30S IC in the early expressome likely results from the long mRNA linker in which the start codon at the P-site of the 30S IC and the 3′-end at the TEC lie almost 90 nucleotides apart. In this regard, the orientation between 30S subunit and TEC is different from that in 70S-containing expressomes with increasing lengths of mRNA linker TTC-A and TTC-B[9–11]. However, it resembles the orientation reported for a TTC-LC assembly containing an extended mRNA linker exceeding the length required for TTC-B formation[16].

Overall, our structural data demonstrates that a putative early expressome can be formed in vitro from a TEC and a 30S IC, and the physical coupling of the two subcomplexes relies on NusG which forms a link between mRNA-engaged 30S and RNAP.

## The coupled transcription elongation-translation initiation complex can be accessed at a target site by sRNA for effector recruitment

The linker mRNA in the early expressome model loops out between the TEC and 30S IC, which may provide an opportunity for sRNA pairing to a target site. To test whether the *ompD* mRNA, when embedded in the TEC-IC, is still accessible for recognition by MicC and deactivating cleavage by the RNA degradosome, we performed degradation reactions with the RNA degradosome in the presence of MicC and Hfq (Fig. 4 and Suppl. Fig. 8). We also subjected to degradation reactions the free *ompD* segment as well as the individual sub-assemblies *ompD$_{89}$*-TEC and *ompD$_{89}$*-30S IC. The degradation rates of *ompD$_{89}$* decreased with rising complexity of the assembly, down to a fourfold slower degradation of NusG-containing *ompD*-TEC-IC compared to free *ompD* (Fig. 4B, C). Double-exponential equation fitting of the formation of the +83-cleavage intermediate and degradation of *ompD* is shown in Fig. 4B, C. The formation rate allows comparison of cleavage specificity for the +83 site which is expressed as the formation rate of the +83 intermediate over the degradation rate of the *ompD$_{89}$* starting material (from Fig. 4C). Cleavage specificities for the +83 site in the different *ompD$_{89}$* assemblies are shown in Fig. 4D. The RNA degradosome shows the lowest preference for +83-cleavage on free *ompD* fragment and the highest on the *ompD$_{89}$*-TEC-IC substrate. The fraction of the unspecific +72-cleavage intermediates on the other hand is highest in the case of free *ompD* fragment, reduced for the *ompD$_{89}$*-TEC and -IC and was unquantifiable in the early expressome (Suppl. Fig. 8). The observation that MicC-guided processing by RNase E occurs with higher specificity when RNAP and/or 30S IC are engaged with the *ompD* mRNA suggests that the gene expression machineries might facilitate the recognition of *ompD* by the sRNA and/or present the MicC-*ompD* hybrid in a way that increases susceptibility to attacks by the RNA degradosome at the +83-cleavage site.

The cryo-EM maps of the *ompD$_{89}$*-TEC-IC complex reveal conformational sub-states that may facilitate regulatory access of the sRNA to the seed pairing site in the transcript (Fig. 4E, F). One of the modes of conformational variation is in a relative tilting of the RNAP with respect the 30S IC, and this is correlated with additional density on the exposed segment of the mRNA (right panel, Fig. 4E). The density is not sufficiently resolved to build a detailed model, but the shape of the envelope can accommodate Hfq and may represent a transient species in which the sRNA/Hfq interacts with the exposed seed region. The schematic in Fig. 4F illustrates a potential model for this transient access.

## Discussion

In this study, we have explored the accessibility of a small RNA pairing site in a target transcript at early stages of transcription elongation and translation initiation. We reconstituted TEC mimicking a state of promoter-proximal transcription pausing of the target mRNA and assembled the 30S translation initiation complex (30S IC) on those mRNAs. Cryo-EM single-particle analysis shows that the transcription factor NusG can physically link the 30S IC and the paused TEC, despite the lengthy mRNA linker (> 70 nts) and the in vitro concentrations used that are far below the proposed dissociation constant of the

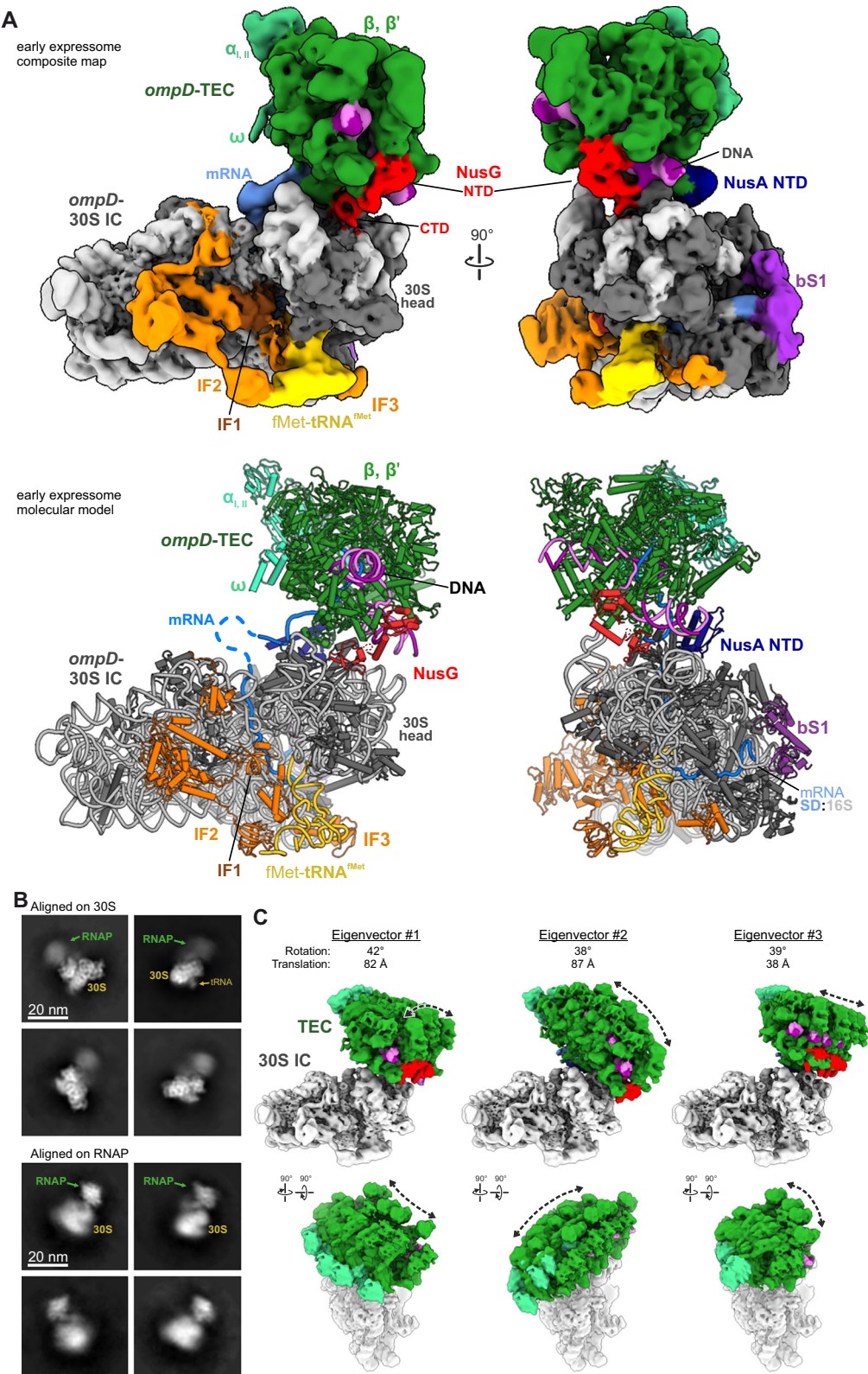

**Fig. 3 | The structure of the transcription elongation-translation initiation complex with regulatory element in the bridging mRNA. A** Composite cryo-EM map (top) and molecular model (bottom) of the "early expressome", comprising NusG-coupled transcription elongation (*ompD*-TEC) and translation initiation (*ompD*-30S IC) complexes. Components that correspond to segments of the map are indicated. The lower panel shows the molecular model interpretation of the cryo-EM structure based on reported structures for the 30S translation initiation complex (PDB: 5lmv) and the NusG-coupled expressome (PDB: 6xgf)[10,33]. **B** 2D class averages of cryo-EM particles that were aligned on either 30S IC or TEC components of the early expressome assembly. **C** Multi-body motion between TEC and 30S IC is visualized by 4 superimposed maps from each of the three main eigenvectors. Maximum changes in angles of rotation and distance between mass centers are indicated.

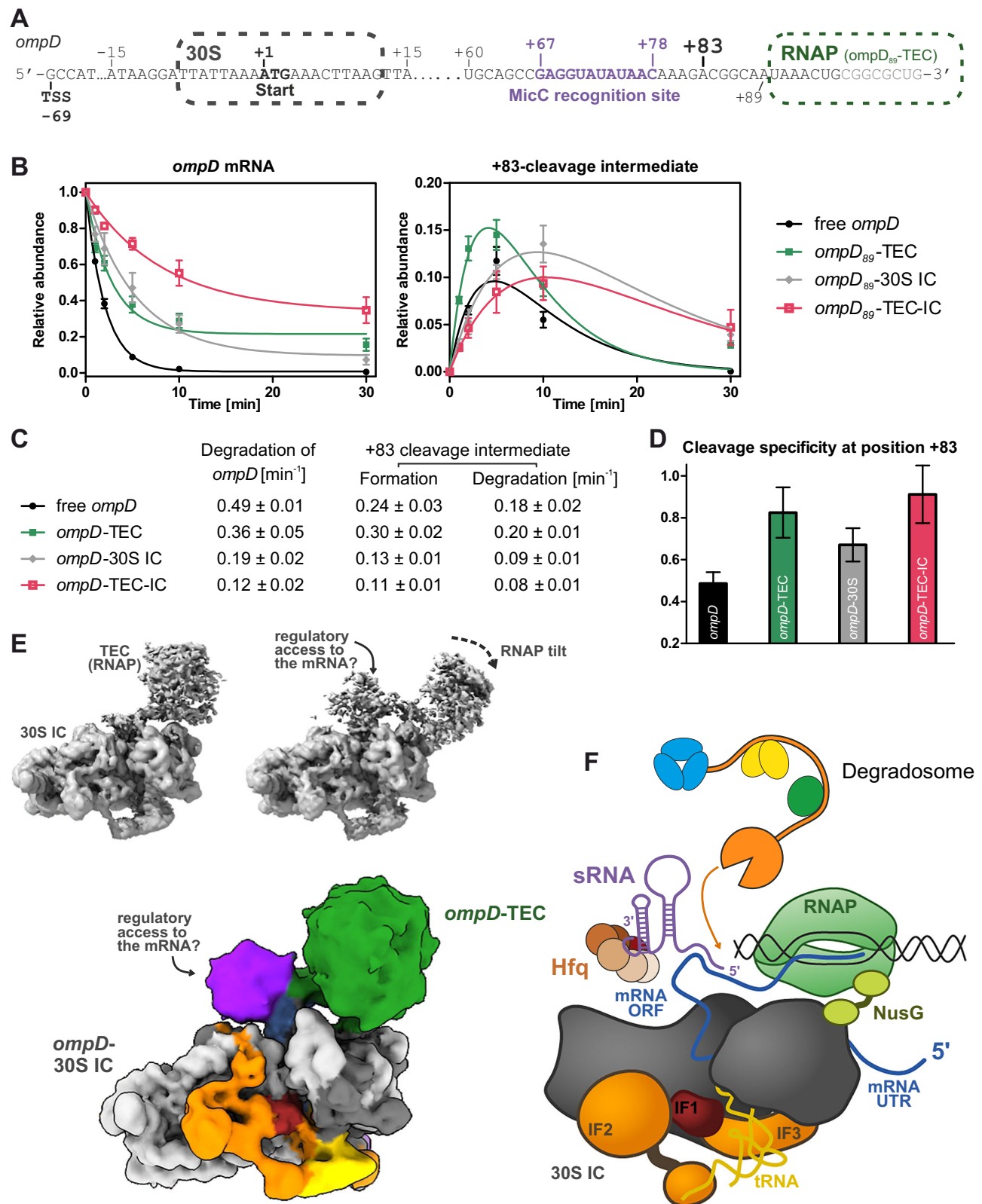

**Fig.** (B) *ompD* mRNA; +83-cleavage intermediate — free *ompD*, *ompD*$_{89}$-TEC, *ompD*$_{89}$-30S IC, *ompD*$_{89}$-TEC-IC.

(C)

| | Degradation of *ompD* [min$^{-1}$] | +83 cleavage intermediate | |
| --- | --- | --- | --- |
| | | Formation | Degradation [min$^{-1}$] |
| free *ompD* | 0.49 ± 0.01 | 0.24 ± 0.03 | 0.18 ± 0.02 |
| *ompD*-TEC | 0.36 ± 0.05 | 0.30 ± 0.02 | 0.20 ± 0.01 |
| *ompD*-30S IC | 0.19 ± 0.02 | 0.13 ± 0.01 | 0.09 ± 0.01 |
| *ompD*-TEC-IC | 0.12 ± 0.02 | 0.11 ± 0.01 | 0.08 ± 0.01 |

(D) Cleavage specificity at position +83

NusE-NusG interaction[29]. The physically coupled assembly, which we refer to as an "early expressome" may represent a transient precursor to mature expressomes in vivo. The complex is highly dynamic but has similar orientation between the 30S subunit and TEC as seen in the TTC with the long-range coupling (TTC-LC)[16].

During the preparation of this manuscript, an independent study reported the in vitro reconstitution of transcription elongation-translation pre-initiation complexes[28]. Webster et al. present structures of mRNA delivery complexes between a paused TEC and 30S ribosomal subunit in the pre-active state and propose two pathways that initiate the coupling between transcription and translation through transcription-assisted recruitment of mRNA to the ribosome[28]. One pathway involves ribosomal protein bS1 of the 30S subunit interacting with the TEC and binding the nascent mRNA to

**Fig. 4 | The bridging element in the TEC-IC complex is accessible to the RNA degradosome. A** Schematic of the *ompD* transcript and MicC recognition site. **B**, **C** Cleavage time courses of *ompD* mRNA fragment and +83-cleavage intermediate species (see Fig. 2D and Suppl. Fig. 8 for representative raw data). Free *ompD* mRNA fragment (black), *ompD*$_{89}$-TEC (green), *ompD*$_{89}$-30S IC (grey) and *ompD*$_{89}$-TEC-IC (red) were incubated with MicC/Hfq and then subjected to cleavage by the RNA degradosome. Reactions were stopped at indicated time points and analysed by denaturing PAGE. Data are mean ± SD from three independent reactions. Degradation trajectories of *ompD* were fitted to one phase exponential decay. Formation and decay of intermediates from *ompD*-cleavage at position +83 were fitted to two-exponential equation $[I = \exp(-kt) - \exp(-jt)]$ with degradation rate $k$ and formation rate $j$. **D** Cleavage specificities for nucleolytic processing of site +83 in *ompD* in the different assembly states, expressed as the formation rate of the +83 intermediate over the degradation rate of the respective *ompD* fragment starting material. Data are mean ± SD from three independent reactions. **E** Conformational substates of the TEC-30S IC assembly. **F** Substate of the early expressome proposed to facilitate the accommodation of Hfq/MicC on the bound *ompD* mRNA. The colours of the segmented map in (**E**) correspond to the components shown in the schematic.

form an intermediate or standby complex that directs the nascent transcript to the Shine-Dalgarno interaction site of the 30S subunit. The RNAP is transiently located near the cluster of bS1 OB-domains and mRNA exit site of the 30S and is subsequently repositioned to be near the mRNA entry site in the translational 30S IC. The second pathway involves RNAP directly binding near the mRNA entry site of the inactive 30S, which then transitions to the active form in the assembled 30S IC by folding the 16S rRNA helix 44 away from the mRNA exit channel into its active position on the 30S body[28]. Our data include IF1, IF2 and IF3 and present the 30S IC in "accommodated" active states, which can be formed via both proposed pathways.

The structural data presented here also corroborate how the ribosomal bS1 might be poised to facilitate interactions with 5′-UTR elements. Webster et al. report that the bS1 protein not only facilitates mRNA delivery to selectively accelerate duplex formation with the anti-Shine-Dalgarno sequence but helps RNAP to stimulate translation initiation. bS1 assists the remodelling of mRNA secondary structures in vitro to aid the 30S with mRNA loading and 30S IC formation[49,50]. When the *ompD* mRNA is recruited to the 30S IC, RNase E relies on the assembly of the RNA degradosome for access and efficient MicC-guided degradation of the *ompD* transcript[32]. Ribosomal bS1 has been shown to form direct interaction with several sRNAs from *E. coli*[51], thereby possibly exerting the 30S IC's influence on Hfq-MicC-guided RNase E attack while located on the 30S ribosome. Further, direct interactions with the TEC[52] might also facilitate sRNA-mediated regulation at the early stage of nascent transcript recruitment to the translation machinery. The paused RNAP may increase the recognition efficiency of the target element within *ompD* by Hfq-MicC, consistent with our observation that *ompD*-cleavage by RNase E is more site-specific at the paused TEC. This is supported by a recent single-molecule study showing that co-transcriptional target recognition by sRNA and Hfq close to the RNAP exit channel is more efficient than recognition post-transcription[53]. The biogenesis of 3′-UTR-derived sRNAs that autoregulate their genes[54], along with kinetic modelling studies, provide further support for mechanisms of co-transcriptional regulation by sRNAs[55]. Access to decay machinery during transcription is also supported by observations of co-transcriptional stabilisation of riboswitches against ribonuclease action[56]. Moreover, a recent study presents a model of co-transcriptional endonucleolytic cleavage of RNA in archaea that leads to transcription termination[57], suggesting the possibility of a functionally analogous system in that domain of life. In an early expressome of the *ompD* transcript in which the RNAP is paused at position +106, the deactivating cleavage site at position +83 for RNase E that is preceded by the recognition site for the MicC sRNA, remains accessible to regulatory RNA and deactivating cleavage by the RNA degradosome in in vitro reactions. Nevertheless, the relationship between the kinetics of sRNA-mediated regulation and the rate of TTC remains unclear, leaving open the question of how much of a temporal window of opportunity is available for sRNAs to access their targets. In addition, the influence of spacer length in long-range coupling on target accessibility to sRNAs remains to be elucidated.

The cellular location of the degradosome must be considered to explain how access might be gained to the putative TEC-30S IC complex tagged with a small RNA (Fig. 5). The envisaged recognition event, in which sRNA recruits degradosome to the 30S IC or TEC-IC complex, could occur in species such as *Caulobacter crescentus*, in which the RNA degradative machinery can be associated with the nucleoid or ribonucleoprotein condensates in the cell interior and could have access to nascent mRNA targets[58]. However, in species such as *E. coli* and *Salmonella* where the degradosome is compartmentalized to the cytoplasmic membrane, and potentially distant from the nucleoid[59], the sRNA-tagged early expressome might be the target for other effectors, such as Rho RNA translocase that terminates transcription through allosteric manipulation of the elongation complex[15,55,60,61]. Rho can act to trigger premature transcription termination in the absence of TTC[58,62] and its activity on nascent mRNA can be modulated by sRNA[63]. Transcription-terminated transcripts could then encounter the degradosome on the membrane upon their diffusion following the release from RNAP.

Another scenario in which membrane-bound RNase E can act early on nascent transcripts is if the transcriptional-translational machinery is brought close to the membrane, as occurs in the transertion mechanism which was described for *Vibrio parahaemolyticus* and suggested to occur also in *E. coli*[64−69]. Transertion, a coupling of transcription, translation and membrane insertion at the membrane component of the type III secretion system, is a process suggested to be common for bacterial membrane proteins. Evidence indicates that membrane-associated RNase E can act on transcripts encoding membrane proteins in *E. coli* to result in co-transcriptional degradation[58].

A fourth scenario in which we envisage RNase E to act on a mRNA-30S IC complex is if the leading edge of a polyribosome comes into proximity of the membrane. Earlier studies have shown that RNase E and the degradosome can interact with polyribosomes in vitro and potentially in vivo, and it was proposed that this forms a passive complex that does not cleave the RNA until activated by a signal, such as a cognate sRNA[70]. In species without RNase E, similar processes are likely to occur, such as in the model firmicute species *Bacillus subtilis*, where the membrane-bound degradosome is based on the distinct ribonuclease RNase Y[71]. This mode of degradation would enable the co-translational decay of a transcript. It also represents a more economical means of exiting translation, because it avoids generating transcript fragments that lack stop codons and entail the hidden metabolic costs of using the tmRNA system for rescue and recovery. The above mechanisms could account for processes of co-translational decay[72,73] analogous to the recruitment of the 5′-to-3′ exoribonuclease Xrn1, which follows the terminal translating ribosome identified in yeast and other eukaryotic species[74,75].

## Methods
### Protein production
Wild-type 30S subunits were prepared from *E. coli* MRE600 strain using zonal centrifugation[76,77]. Initiator tRNA (fMet-tRNAfMet) and *E. coli* initiation factors IF1, IF2 and IF3 were purified according to published protocols[78]. *E. coli* Hfq was purified as described[79]. The RNA degradosome was prepared as described[80]. *E. coli* core RNAP was expressed from pVS10 plasmid coding for all five subunits[81] and purified as described[82].

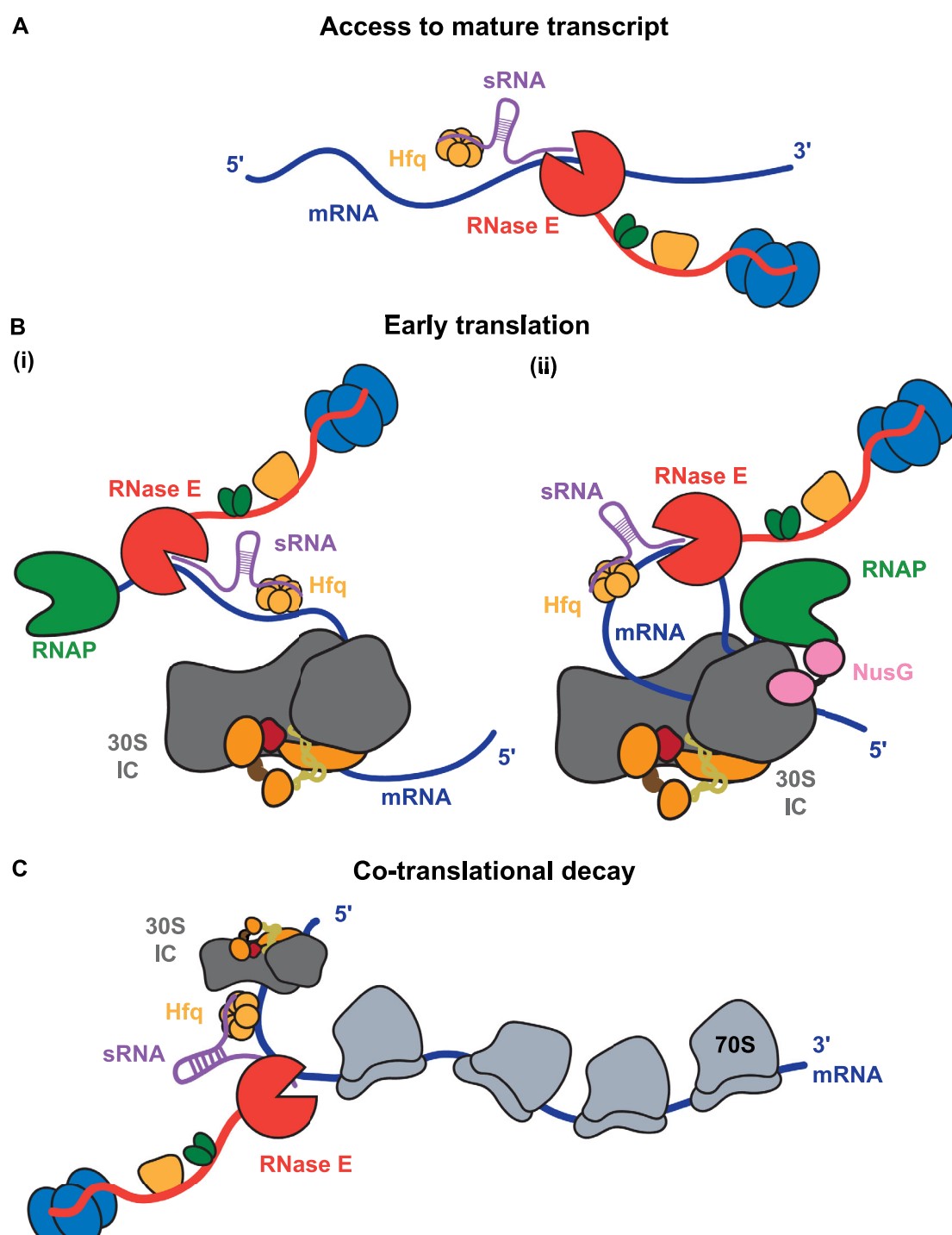

**Fig. 5 | Schematic models for sRNA-mediated degradation of mRNAs during transcription and translation.** Four modes are presented where the RNA degradosome could access transcripts tagged with sRNAs. **A** Access to the mature mRNA after transcription termination. **B** Access at early stages of translation explored in this study. Mode (i) depicts a scenario in which the degradosome can access the coding region following the formation of the pioneering 30S IC. Mode (ii) posits access to the translation initiation complex in the vicinity of the paused transcription elongation complex. **C** Access for co-translation degradation. Here, the degradosome could be interacting with polysomes once the mRNA has begun to be translated. This mode is supported by the in vivo observation of RNA cluster formation by degradosomes in the presence of polysomes (Hamouche et al. 2021).

## RNA production

RNAs were prepared by in vitro transcription (IVT) as described[31]. In brief, IVT templates were amplified by PCR from plasmid pVP042-3 that carries the *Salmonella* Typhimurium *ompD* gene[30]. The forward primer introduces a T7 promoter sequence upstream of the transcription initiation site and, where indicated, the reverse primer encodes a non-canonical 9 nts sequence (CGGCGCUGG) at the 3' end of the corresponding transcript. IVT reactions containing 3 µg of template DNA, 5 mM of each of ATP, UTP, GTP and CTP, 10 mM DTT and 0.5 U/µL RNaseOUT™ (Invitrogen) were incubated with recombinant T7 RNAP in 40 mM Tris, pH 8.0, 25 mM MgCl$_2$, 2 mM spermidine at 37 °C for 5 h. IVT products were DNase I-treated, resolved by 4%

denaturing PAGE and excised RNA bands were electroeluted using Elutrap™ Electroelution System Kit (Whatman), followed by clean-up using PureLink™ RNA Micro Kit (Invitrogen).

## Assembly of transcription elongation complexes

TECs were assembled following a previously described strategy[60]. To anneal nucleic acid scaffolds, 8 μM mRNA fragment was mixed with template DNA (tDNA) in a 1:1.25 molar ratio in 10 mM TRIS, pH 7.6, 40 mM KCl, 5 mM MgCl₂, heated to 95 °C for 2 min and slowly equilibrated to 37 °C. Two volumes TEC reconstitution buffer (20 mM TRIS, pH 7.6, 120 mM potassium acetate, 5 mM magnesium acetate, 1 mM TCEP) were pre-heated to 37 °C and added to annealed mRNA/tDNA. *E. coli* RNAP was added in 1:1 ratio to RNA, followed by 10 min incubation. Non-template DNA (ntDNA) was added in 1:1 ratio to tDNA and followed by incubation for 10 min at 37 °C. Assembled TECs were prepared fresh and used immediately for degradation assays or assembly of higher order complexes.

For cryo-EM specimen preparation (Sample 1), 5 μM TEC was mixed with equimolar amount of pre-incubated (32 °C for 30 min) MicC and Hfq injected onto a 3.2/300 Superose 6 size-exclusion chromatography column equilibrated with 20 mM TRIS-Cl, pH 7.6, 120 mM potassium acetate, 5 mM magnesium acetate, 2 mM DTT, 10 μM ZnCl₂. Fractions containing all components as determined by urea-PAGE and SDS-PAGE were pooled, concentrated to 40 μL and used directly for vitrification on graphene oxide coated gids as described below.

For Sample 2, 7.5 μM TEC was mixed with 1.4-fold molar excess of pre-incubated (32 °C for 10 min) MicC and Hfq and dialysed against TEC reconstitution buffer to remove residual glycerol. The reaction was recovered and concentrated to approx. 10 μM using a 30 MWCO spin concentrator. 8 mM CHAPSO detergent were added and 3 μL sample was applied to both faces of an R2/2 UltrAuFoil grids and vitrified after 2 s blotting on both sides.

## Preparation of *ompD*−30S IC and early expressomes

30S IC was prepared following a previously described strategy[83]. In brief, 30S subunits were incubated in buffer TAKM₂₀ (50 mM Tris-HCl [pH 7.5], 70 mM NH₄Cl, 30 mM KCl, 20 mM MgCl₂) for 30 min at 37 °C for reactivation. Reactivated 30S subunits were incubated with a 2.5-fold excess of mRNA, twofold excess of initiation factors 1−3 and a 2.5-fold excess of initiator fMet-tRNA^fMet in the presence of 250 μM GTP or GTPγS (Jena Bioscience) in TAKM₇ buffer[83]. For transcription/translation-coupled complexes, the 30S IC was prepared as described above using pre-assembled *ompD*-TEC instead of free mRNA and omitting the chromatography step. Coupled TEC-IC for cryo-EM analysis was prepared with final concentrations 0.3 μM 30S, 1.2 μM TEC, 0.6 μM of IFs and tRNA, 10 μM NusA, 20 μM NusG and 5 μM MicC/Hfq and directly used for grid preparation on Quantifoil 1.2/1.3 grids with a 2 nm amorphous carbon layer as described below.

## In vitro transcription

The first 306 bases of the *ompD* gene (−69 to +237) were inserted into the pMMB67HE vector between *tac* promoter and T1 terminator[84]. For IVT assays, 15 μL reactions containing 5 nM *ompD*-pMMB67HE, 0.5 U/μL RNaseOUT™ and 0.5 U *E. coli* RNAP Holoenzyme (NEB, M0551S) were pre-incubated in reaction buffer (40 mM TRIS, pH 7.5, 150 mM KCl, 10 mM MgCl₂, 1 mM DTT, 0.01 % Triton ×-100™) at 37 °C for 5 min. Reactions were started by adding ribonucleotide mix to 5 mM, incubated at 37 °C for 1 h and quenched by addition of equal volume stop buffer (200 mM TRIS, pH 7.5, 25 mM EDTA, 300 mM NaCl, 2% SDS, 0.5 mg/mL Proteinase K). After proteolysis at 50 °C for 1 h, samples were mixed with 0.5 volumes 2× RNA Loading Dye (Thermo Scientific), heated to 95 °C for 2 min and separated by urea PAGE on 8 % polyacrylamide gels.

## In vitro RNA degradation assays

Degradation assays were carried out following a protocol described in ref. 31. Degradation reactions were performed in degradation buffer (25 mM TRIS, pH 7.6, 50 mM NaCl, 50 mM KCl, 10 mM MgCl₂, 2 mM DTT) with 0.5 U/μL RNaseOUT™. 5′-monophosphorylated MicC sRNA was brought to 50 °C for 2 min and slowly cooled to 37 °C before use. Substrates (*ompD*-TEC, *ompD*-TEC-30S IC or *ompD*:tDNA:ntDNA for control) were prepared as described above and pre-incubated with equimolar amounts of MicC and Hfq at 37 °C for 15 min at final concentrations of 0.2 μM. After withdrawal of reaction aliquots for $t = 0$ min, fresh RNase E^{1–598} or full-length RNA degradosome was diluted in 2× degradation buffer. Reactions were started by adding 1/10 volume of enzyme at final concentrations of 0.05 μM for RNase E^{1–598} or 0.01 μM degradosome. Reaction aliquots were withdrawn at indicated time points and quenched with equal volume stop buffer (see above under IVT) at 50 °C for 30–45 min. Samples were mixed with RNA loading dye II (ThermoFisher), heated to 95 °C for 2 min and resolved on 8% or 10 % denaturing (7.5 M urea) polyacrylamide gels. Gels were stained in SYBR Gold (Invitrogen) and visualized under UV light. Bands were quantified with ImageJ and plotted with GraphPad PRISM as fraction of $t = 0$ min. Degradation rates of full-length RNAs was determined by fitting the data to the one phase decay fit equation in GraphPad PRISM, $[I = (IO - P) * e\text{-}kt) + P]$ with relative intensity, $I$; plateau, $P$; degradation rate, $k$; reaction time, $t$. Formation and decay rates of intermediates from *ompD*-cleavage at position +83 were determined by fit to two-exponential equation $[I = \exp(\text{-}kt) − \exp(\text{-}jt)]$ with degradation rate $k$ and formation rate $j$. Cleavage site specificities were calculated by dividing intermediate production rates by the respective full-length RNA degradation rate.

## Graphene oxide coating and grid preparation

Graphene oxide (GO) coating was prepared by an adaptation of the drop-casting method[85]. Quantifoil Cu 300 1.2/1.3 grids were glow-discharged on the darker carbon side (PELCO easiGLOW: 15 mA, 0.28 mBar, 2 min). GO solution (Sigma-Aldrich 763705, 2 mg/mL dispersion) was diluted tenfold in water and centrifuged at $300 \times g$ for 30 s to pellet insoluble GO flakes. The supernatant was further diluted tenfold to a working concentration of 0.02 mg/mL. 1 μL working solution was applied to the glow-discharged side of the Quantifoil grids and dried at room temperature for 10 min. GO-coated grids were kept at room temperature for at least 16 h and then directly used for sample vitrification. Quantifoil 1.2/1.3 grids with a 2 nm amorphous carbon support layer were briefly glow-discharged (PELCO easiGLOW: 10 s, 25 mA, 0.39 mBar) directly before sample application. For vitrification, 4 μL of sample was applied to grids and incubated for 30 s. Using a FEI Vitrobot, excess sample was blotted and grids were plunge-frozen in liquid ethane. Grids were subsequently stored in liquid nitrogen until screening and collection.

## Cryo-EM data collection and processing

For structures of *ompD*−30S IC, 9104 multi-frame movies were collected on a 300 kV FEI Titan Krios equipped with a Gatan K3 detector. For the TEC structure, a dataset of 1619 multi-frame movies was collected on a 300 kV Titan Krios equipped with a Falcon 3 detector in Counting mode. For structures of different TEC-30S IC reconstitutions, small datasets were first collected on a 200 kV Talos Arctica equipped with a Falcon 3 detector in Counting mode and processed in Relion 3.1[86] as depicted in image processing workflows (Suppl. Fig. 7B, G, L). For the coupled TEC-30S IC structure, 12,011 multi-frame movies were then collected on a 300 kV FEI Titan Krios equipped with a Gatan K3 detector. Details for data collection are summarised in Supplementary Table 1. Multi-frame movies were motion-corrected in Relion 3.1[86] and the generated micrographs were imported in cryoSPARC v4[87] and further processed as follows and illustrated in Suppl. Figs. 1−3. Particles were picked using "Blob Picker", extracted and downsampled fourfold.

For all described structures, an initial round of Heterogeneous Refinement was run on all extracted particles using 3D references generated by ab initio reconstruction from particles subsets selected after 2D classification to isolate 30S or TEC-containing particles from poor-quality "junk" particles.

For 30S structures, a subsequent 3D classification without refining particle alignments separated particles containing tRNA from particles without tRNA. tRNA-containing particles were further classified with focused masks around IFs and tRNA to isolate IF2-containing particles, which were re-extracted without downsampling, subjected to Homogenous Refinement, and further classified with a focused mask around IF2 to isolate particles that refine to highest resolution in this region. Separately, particles lacking tRNA were classified iteratively with focus masks around IFs and mRNA and classes with weak or no density for these subunits were discarded as illustrated in Suppl. Fig. 1B. The final particle set was re-extracted without downsampling.

For the *ompD*-TEC structure, particles isolated by heterogeneous refinement, as described above, were extracted without downsampling and subjected to global CTF refinement and one round of a 3D classification, from which the class with the particles that refined to the highest resolution was kept. Final particle sets were subjected to Homogenous Refinement with default settings and resulting reconstructions were subjected to Local Resolution Estimation and Local Filtering. Particle orientation plots and local resolution estimates are shown in Suppl. Fig. 6. 3D Variability Analysis (3DVA)[88] was performed on final particle sets of the 30S structures and visualised in "Simple" mode. For the TEC, 3DVA was performed on all TEC particles that refined to better than 4 Å resolution to better reflect the variability across the sample.

For the coupled TEC-30S IC structure, all non-junk particles that aligned on the 30S reference were subjected to a second round of Heterogeneous Refinement, and the particle class containing density for IFs and tRNA in the 30S IC was selected. A soft-edged mask around the 30S IC was used to subtract the signal of the 30S IC followed by 3D classification without re-alignment. The classes with the strongest density in the region of the TEC were selected and subjected to Heterogeneous Refinement providing low-pass filtered (20 Å) reference maps of the TEC to isolate particles that can be aligned on RNAP (these 34,106 particles were processed further; see below). After a Local Refinement of the best class using a Mask around the TEC, a final round of 3D classification was run, particles of the best class were re-extracted with twofold downsampling and locally refined (Suppl. Fig. 5C "TEC portion") or subjected to Homogenous refinement providing the 30S IC reference (Suppl. Fig. 5C "Consensus Refinement"). Using the particle alignments of the local refinement around the TEC, the TEC signal was subtracted from the re-extracted particles and followed by a Homogenous Refinement (Suppl. Fig. 5C "30S IC portion"), which was combined with the TEC portion into a composite map inside ChimeraX. The final set of expressome particles was converted to a STAR file using UCSF PyEM (https://doi.org/10.5281/zenodo.3576630), imported into Relion and a subjected to a Multi-body Refinement[48] providing a local mask around 30S IC and TEC, respectively. Multi-body motion along the three main eigenvectors is visualised in Fig. 3C and Suppl. Movie 2. Angles of rotation and change of distance between mass centres were calculated inside PyMOL using RotationAxis (http://pymolwiki.org/index.php/RotationAxis).

Separately, the subset of 34,106 expressome particles (see above) was classified based on the position of the TEC in respect to the 30S IC by using the 30S-subtracted particles that contain signal for the TEC while providing alignments of the original particles that were refined on the 30S IC. The two classes that displayed additional density around the path that the mRNA would take between TEC and 30S IC were classified with a focus mask drawn around this additional density. The

two classes with the strongest density in this area were refined individually by Homogenous Refinement (Suppl. Fig. 5E).

## Data availability
The data supporting the findings of this study are available from the corresponding authors upon request. Structure data are available from the EMDB.

EMD-55527 Early Expressome Composite Map of TEC and 30S IC
EMD-55528 Early Expressome Consensus Refinement
EMD-55529 Early Expressome RNAP/TEC body
EMD-55530 *E. coli* 30S IC containing mRNA, IF1 and IF3
EMD-55531 *E. coli* 30S IC containing mRNA, initiator tRNA, IF1 and IF2

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

## Acknowledgements

We thank Andrzej Szewczak-Harris and the staff of the Cryo-EM Facility, Dimitri Y. Chirgadze, Steven Hardwick and Lee Cooper, for assistance with data collection. We thank the staff at eBIC for access to facilities and help with data collection. We thank Tom Dendooven, Kathi Frohlich and Joerg Vogel for helpful comments and suggestions. We thank Kai Katsuya Gaviria for providing Hfq and helpful advice. K.J.B., B.F.L. and N.Z. are supported by Wellcome Trust Investigator Awards (222451/Z/21/Z, 200873/Z/16/Z, 217189/Z/19/Z). J.J.R. is supported by a Herchel Smith Studentship. G.P. was supported by a Benn W Levy–Vice Chancellor Award SBS DTP PhD studentship.

## Author contributions

All authors contributed to data collection. CryoEM data and analysis were done by J.R. and G.P. All authors contributed to writing the manuscript.

## Competing interests

The authors declare no competing interests.
