## [Transparent Peer Review file · Nature Communications]

Structure of the 30S translation initiation complex coupled to paused RNA polymerase and its potential for riboregulation

Corresponding Author: Professor Ben Luisi

Version 0:

Reviewer comments:

Reviewer #1

(Remarks to the Author)

The current view is that transcription and translation are coupled in many prokaryotic species. Furthermore, the cooperation between RNA polymerase (RNAP) and the ribosome may have functional consequences that are difficult to predict. However, the steps involved to establish coupling between RNAP and the pioneering ribosome are still poorly understood. This manuscript by Roske, Paris, et al. describes cryo-EM reconstructions of *E. coli* small ribosomal subunit initiation complexes (30S IC) and a paused RNAP and explores how the *ompD* transcript is regulated via degradation by a small regulatory RNA (sRNA) called MicC. The authors describe reconstructions of two 30S ICs in isolation, an RNAP transcription elongation complex (TEC) assembled on the putative *ompD* pause site, as well as complexes of the TEC coupled to the 30S IC via the transcription factor NusG (and potentially also NusA). Overall the manuscript is well written but I have several concerns and some aspects would benefit from clarifications and additional experimental evidence (see below) before publication can be considered.

Major comments:

An important concern is that the authors decided to assemble a TEC at two different positions with respect to the MicC binding site to test RNA degradation (OmpD99 and OmpD105). If I understand correctly, the TEC at *ompD105* was then also used for the complex between 30S-IC and the TEC. All of this relies on the fact that the *ompD105* TEC recapitulates the paused RNAP but this is where I am skeptical.

1. The authors decided to alter the RNA/DNA hybrid sequence compared to the wildtype situation (see their supplementary Figure 2F, and 2G). As pointed out by the authors themselves, the RNA 3'-end, and the next template and/or non-template DNA base are believed to be critical to induce pausing in a TEC (see their supplementary Figure 2A). This reviewer is puzzled why the authors decided against using the natural sequence, which supposedly induces a paused TEC conformation. The concern is of course that the TEC is not adopting a paused conformation and thus skewing the results.
2. Further, the *ompD*-TEC sequence as depicted in Figure 2D does not match the one depicted in supplementary Figure 2F and neither of them matches the wildtype sequence of the putative pause site as depicted in supplementary Figure 2B. Please explain.
3. The authors make little comment about the *ompD105*-TEC reconstruction. Does it exhibit any of the characteristics that have been observed in paused TECs (swivelling, RNA/DNA hybrid tilting, more pre- as opposed to post-translocated register)?
4. Why would the *ompD105*-TEC lack omega but not the *ompD105*-TEC-30S-IC – is there any difference in preparation that could explain this discrepancy?

There is very little to no comment on NusA in the reconstructions and biochemistry. Does NusA contact the 30S-IC? If so, is it in a similar fashion to what has been reported for expressomes (see Wang et al., Science 2020).

The authors suggest that the 30S and TEC facilitate recognition of *ompD* transcript for degradation. My first thought was, since degradation rates are reduced in presence of 30S and TEC (Figure 4B and 4C) that the gene expression machinery provides protection – wouldn't that make more sense? As long as the mRNA is transcribed (and more importantly) translated, it is protected from degradation? This agrees with the reduced rate of overall *ompD* degradation and also reduced rate of +83 intermediate degradation, no? Also, it is unclear to this reviewer how specificity for +83 is measured (Figure 4D)?

Minor comments:

The authors mention that the 30S head exhibits less movement in absence of IF3 in the 30S-IC reconstructions – what is the metric for this observation? Did the authors perform 3D variability analysis and compared the extreme head positions? Please clarify in the manuscript.

The authors mention that their bS1 conformation (I suggest to use the revised nomenclature for all ribosomal proteins – i.e. bS1 not S1, uS10 not S10, etc. – see Ban et al., *Current Opinion in Structural Biology*, 2014, PMID 24524803) is different from expressome structures but similar to a recent report of complexes between RNAP and the 30S subunit (Webster et al., *Science* 2024, PMID 39607923). What exactly is different? As far as I know, bS1 has either not been modelled in the available expressome structures or only the N-terminal helix and the first OB domain of bS1 have been deposited. I believe this reflects the fact that bS1 is mobile and only OB1 is in a somewhat stable orientation with respect to the rest of the 30S and is consistent with Webster et al., *Science* 2024.

In that sense, the statement of the authors, which suggests a difference in bS1 conformations between 30S and 70S ribosomes/expressomes is misleading and should be removed or further clarified (what part of bS1 is different and compared to which reconstruction exactly).

Figure 1 shows cryo-EM reconstructions of two 30S populations (30S-IC with IF1 and IF3, and 30S-IC with IF1, IF2, and fMet-tRNA^{fMet} – it is fMet-tRNA^{fMet}, i.e. aminoacylated and formylated initiator tRNA, correct?). Are these consensus reconstructions? If so, the authors may consider to perform focused refinement of 30S head and body to obtain higher local resolution.

As mentioned before, the ompD-TEC cryo-EM reconstruction (Fig. 2A-C) lacks the RNAP omega subunit according to the authors. I noticed that *E. coli* RNAP was expressed from a plasmid that encodes all 5 RNAP subunits but no additional plasmid was used to express the omega subunit. I seem to remember anecdotal evidence that rpoZ needs to be overexpressed from an additional plasmid to ensure stoichiometric amounts and was curious if the authors have considered this to explain their observation.

Supplementary Figure 1C: the legend still contains a comment from editing and the map is segmented but not coloured by local resolution as stated in the legend.

Supplementary Figure 1D: authors should include particle orientation plot

Supplementary Figure 1E: the map is segmented but not coloured by local resolution as stated in the legend.

Supplementary Figure 1F: authors should include particle orientation plot

Supplementary Figure 2C-2D: The pause signal is very weak and the authors may miss the time-point when the paused species reaches its maximum. A time course would be much more useful to appreciate accumulation of paused species and decay as RNAP escapes from the pause. There is also no obvious effect of NusA and/or of temperature on the extent of pausing – again, a time course (i.e. RNA transcription kinetics) would reveal an effect. In S2D it is unclear which lane was used to generate the profile – is this an average of all 4 lanes? That's not very useful in my opinion. In general, a radioactive (³²P) assay may be superior to staining with SYBR gold. It is also unclear to me why the authors decided to flank the ompD sequence with vector-derived sequence? Why not use the regular ompD sequence – this would avoid any potential effect of upstream sequence on pausing and would also make analysis easier because the paused species is shorter, easier to resolve, and its length could be more reliably measured on high-resolution denaturing PAGE. This comment also reflects another concern I highlighted earlier: the fact the authors use inconsistent sequences is a problem in my opinion: i) the sequence for the assays is different from the sequence used for the cryo-EM reconstructions; and ii) the terminal 9 nucleotides used for cryo-EM reconstructions does not reflect the ompD pause sequence.

Supplementary Figure 2M,N: The legend says ompD105 but the figure says ompD99. The bottom plot should presumably say ompD83 not ompD99? Also, the authors propose that the TEC protects ompD99 from cleavage mediated by MicC yet ompD99 degradation is not affected according to the plot in S2N – please elaborate.

Further, I think it would be fair to include ompD RNA alone as a negative control so we can appreciate that the RNA is stable in absence of RNase E under these reaction conditions. Is the RNA quantified as fraction of total signal in each lane?

I would encourage the authors use the number of nucleotides rather than codons to describe the distance between RNAP and the ribosomal P-site because RNAP moves in steps of single nucleotides and there may not be an integral number of codons them.

Reviewer #2

(Remarks to the Author)

The authors report a cryo-EM structure of an *E. coli* transcription-translation complex (TTC) comprising a promoter-proximally paused *E. coli* transcription elongation complex (TEC) and a ribosome 30S subunit translation initiation complex, coupled by mutual interaction with the transcription elongation factors NusA and NusG.

The authors further show that, in this complex, the mRNA spacer between the TEC and the ribosome 30S subunit is accessible to interactions with a small regulatory RNA and to cleavage by the RNA degradosome assembly.

The authors infer that physical coupling between transcription and translation can be established at the initiation stage of translation and that the mRNA spacers in coupled complexes at the initiation stage of translation are accessible to regulatory factors.

The structural organization of the TEC and ribosome in the authors' complex matches, in detail, the structural organization of the recently reported long-range transcription-translation complex (TTC-LC; <https://www.biorxiv.org/content/10.1101/2024.07.20.604413v1>), and the inferred functional role of the authors' complex and accessibility to regulators of the authors' complex matches the functional role and regulatory accessibility recently proposed for TTC-LC (<https://www.biorxiv.org/content/10.1101/2024.07.20.604413v1>).

The results are important and timely and warrant publication. However, prior to publication, the manuscript must be revised to cite and describe the recent structural work on TTC-LC and to cite and describe the proposed functional role and regulatory accessibility of TTC-LC (<https://www.biorxiv.org/content/10.1101/2024.07.20.604413v1>). The recent structural work on, and proposal about, TTC-LC provide crucial context that makes the authors' work important and timely, but, surprisingly, the current manuscript does not cite or describe the previous work.)

Specific comments

28: Add "and transcription factor NusA" (unless evidence is obtained and presented that NusA does not contribute to stability of the complex).

57: Add citations to <https://pubmed.ncbi.nlm.nih.gov/39117885/> and <https://www.biorxiv.org/content/10.1101/2024.07.20.604413v1>. Delete citation to Qureshi & Duss, 2025 (which does not report a cryo-EM structure).

61: Replace "structural" by "structural and single-molecule"

65-66: Add citations to <https://pubmed.ncbi.nlm.nih.gov/39117885/> and <https://www.biorxiv.org/content/10.1101/2024.07.20.604413v1>.

83: Cite and describe hypothesis that translation initiation and ribosome catch-up involve a long-range, loosely coupled transcription-translation complex (TTC-LC, with a long, >11 codons, looped-out mRNA spacer between TEC and ribosome, and with accessibility of the mRNA spacer to regulatory factors), and that subsequent coupled transcription and translation involves a short-range, tightly coupled transcription-translation complex (TTC-B, with a different orientation of ribosome relative to TEC, with short, 7-11 codons, mRNA spacer, and with little or no accessibility of the mRNA spacer to regulatory factors)(<https://www.biorxiv.org/content/10.1101/2024.07.20.604413v1>).

92: State that the complex obtained has the same orientation of ribosome S30 subunit to TEC as in TTC-LC (<https://www.biorxiv.org/content/10.1101/2024.07.20.604413v1>).

93: Add, "as proposed for TTC-LC in <https://www.biorxiv.org/content/10.1101/2024.07.20.604413v1>."

280-286: Is cryo-EM density for the NusG interdomain linker present? If not, the linker should be omitted--or should be rendered or colored differently--in Fig 3B.

286: Add text describing density and interactions for NusA (which is shown to be in the complex in Fig 3B, but is not discussed in the current text).

289-292: Perform and report results of analogous analysis assessing whether formation of the complex is dependent on NusA.

297-299: Add figure panels to Fig 3 showing superimpositions of the authors' complex on TTC-B and TTC-LC. (Panels A, C, and D of Fig 3 can be moved to the Supplement, if additional space is needed.)

297-299: Replace current text--which is incorrect unless "general" in "general orientation" is interpreted *very* loosely--with text stating that the the orientation of ribosome 30S subunit to TEC is different from that in TTC-B but is the same as that in TTC-LC.

300: Add paragraph on the mRNA spacer between ribosome S30 subunit and TEC. For what part or parts of the mRNA spacer is cryo-EM density detected? What inferences about the path of the spacer mRNA can be made? For example, can it be inferred that the mRNA spacer is too long to follow a straight path and therefore must be looped out?

344-347: Add "and transcription factor NusA" (unless evidence is obtained and presented that NusA does not contribute to stability of the complex).

387-389: State that the complex obtained has the same orientation of ribosome S30 subunit to TEC as in TTC-LC (<https://www.biorxiv.org/content/10.1101/2024.07.20.604413v1>).

391: Delete "similar." Clarify the differences.

Reviewer #3

(Remarks to the Author)

In this work, cryo-EM is used to investigate complexes formed on ompD mRNA, which is subject to regulation by the small RNA, MicC, in Salmonella. Structures of 30S initiation complexes (one with IF1 and IF3; another with IF1, IF2 and fMet-tRNA) were determined, as were structures of the 30S IC and RNAP. For the latter, RNAP was positioned via a heteroduplex (bubble) template ~100 base pairs downstream from the start codon. RNAP and the 30S subunit interacted with one another through NusG / NusE (S10), with mRNA (~70 nt) presumably looped out (unresolved) between the two machines. The MicC recognition site is positioned in the middle of this looped-out mRNA. Treatment of these complexes with RNase E and MicC/Hfq resulted in mRNA decay, the kinetics of which differed depending on which complex was present. While this work provides some impressive structural data, including for example new views of S1, the biological relevance here is unclear and the paper is dominated by speculative text.

Specific concerns:

1. Pfeiffer et al 2009 showed that transplantation of the MicC binding site into another gene (ompN-gfp) leads to MicC-dependent control, akin to that seen for the native gene. This argues against a complicated mechanism. Is there any evidence that MicC-dependent control of ompD requires transcription-translation coupling? Without such evidence, the rationale for this work is unclear.
2. In the RNase E assay, mRNA decay is seen whether it is MicC-guided or not. This makes interpretation of the data tricky, to say the least. The authors state that the "+72 cleavage product is an off-target species only observed in vitro." How then do we (the readers) interpret overall decay rates in these experiments? In the cell, there is a large difference in decay rates (+/- MicC), which this in vitro assay fails to recapitulate.
3. The authors speculate a lot about transcription-translation coupling and its potential role in targeted mRNA decay. Yet, no experiments to disrupt coupling and monitor the consequences in the cell are presented.
4. In the RNAP-NusG-30S complexes, the authors envision that a single mRNA is bound. Has this been verified? Can similar NusG-linked complexes be made with two separate halves of this mRNA? If not, why not? Addressing these questions could increase the impact of this work substantially.
5. Unclear is whether the pause site in ompD was uncovered by genome-wide experimental data or predicted by the consensus motif derived from such data.
6. Figure 5 is complicated and seems more appropriate for a review on RNase E than for this study. Also, the localization of RNase E in E. coli seems at odds with case (i), the case studied here (what?!?). The extended portion of RNase E (purple string) is difficult to distinguish from the mRNA (blue string).
7. In the Intro, the authors state: "Transcription-translation coupling (TTC) appears to occur extensively in representative gram-negative bacterial species,..." However, an analysis of six operons in E. coli showed evidence for transcription-translation coupling in only one operon and only in the presence of a mutation that slowed down RNAP, suggesting that coupling is uncommon and stochastic (PMID 30275301). Moreover, Hwa and co-workers provided compelling evidence that coordination of transcription and translation in E. coli is mediated indirectly via the concentration of ppGpp and aminoacyl-tRNA (PMID 31451774). Yes, several structural studies have shown how NusG can link RNAP and the ribosome. But, how frequently RNAP and the ribosome are coupled in E. coli and what role this physical interaction plays remain open questions.

Version 1:

Reviewer comments:

Reviewer #1

(Remarks to the Author)

I have been asked to comment on the author's response to comments from both reviewer 1 as well as reviewer 3. Reviewer 3 seemed particularly concerned about the overall impact and relevance.

1.: The authors need to clearly specify that the TEC has not been assembled on the natural pause-inducing sequence but instead on a scaffold with a different sequence to stabilize the TEC. The reader needs to be aware of this potential limitation. It is debatable if a TEC on a non-pause-inducing sequence and a paused TEC are the same and I would argue readers should decide themselves - thus claiming that the TEC mimics a paused complex is not accurate in my opinion and should be avoided. However, I understand perfectly well that the state of the TEC (paused or not) may not have any influence on RNA cleavage. This is really a question of using accurate wording to be clear avoid misunderstandings.

2.: Minor comment: I believe the vast majority of studies in the translation, transcription, and coupled translation-transcription field use nucleotides to denote distance along a (shared) mRNA. Recent work by the Duss lab, Weixlbaumer lab, or Zenkin lab, which addressed transcription-translation coupling also used that metric. However, I am sure readers can multiply the number of codons by 3 if the authors prefer to use codons as their preferred unit.

3.: It remains puzzling why the authors decided to use a coupled transcription-translation system to study targeting by MicC because it seems MicC can target the RNA regardless of whether it is free, still bound to a TEC, or bound to a TEC and a 30S-IC? It would thus be useful to emphasize the rationale of the study more as pointed out by reviewer 3.

Additional comments:

The abstract says "...RNA polymerase that is paused proximally to the promoter can associate with the pioneering 30S translation initiation complex (30S IC) through mutual binding of the transcription factors NusG and NusA." However, the authors were unable to resolve NusA bound to the 30S.

Reviewer #2

(Remarks to the Author)

The revised manuscript fails to address key points raised in my previous review:

92: State that the complex obtained has the same orientation of ribosome S30 subunit to TEC as in TTC-LC (<https://www.biorxiv.org/content/10.1101/2024.07.20.604413v1>).

93: Add, "as proposed for TTC-LC in <https://www.biorxiv.org/content/10.1101/2024.07.20.604413v1>."

289-292: Perform and report results of analogous analysis assessing whether formation of the complex is dependent on NusA.

297-299: Add figure panels to Fig 3 showing superimpositions of the authors' complex on TTC-B and TTC-LC.

300: Add paragraph on the mRNA spacer between ribosome S30 subunit and TEC. For what part or parts of the mRNA spacer is cryo-EM density detected? What inferences about the path of the spacer mRNA can be made? For example, can it be inferred that the mRNA spacer is too long to follow a straight path and therefore must be looped out?

387-389: State that the complex obtained has the same orientation of ribosome S30 subunit to TEC as in TTC-LC (<https://www.biorxiv.org/content/10.1101/2024.07.20.604413v1>).

Prior to possible publication, these key points will need to be addressed.

In addition, the current Abstract (which fails to state the results and conclusions of the work) will need to be revised, and the current text at 527-528 (which misrepresents a hypothesis as a fact and fails to state the rationale for the hypothesis) will need to be revised.

Reviewer #4

(Remarks to the Author)

This manuscript addresses important questions in bacterial RNA biology, including transcription-translation coupling, sRNA regulation, and RNA degradation. They used cryo-EM approaches alongside biochemical assays. Below, I summarize the main contributions of the study as well as my reservations.

- Structural evidence of an early expressome, composed of RNAP and 30S IC (+/- NusG). While this finding is of interest, it is not entirely novel. A similar complex was previously resolved and reported by Webster et al. (Science, 2024). The structural details are beyond my expertise, but the degree of novelty relative to that prior work appears limited.

- sRNA binding within the early expressome. This extends the authors' prior work (Bandyra et al., NAR 2024), which already provided a thorough examination of RNA degradation kinetics in the presence of MicC and the 30S IC in vitro. The present study adds the transcription elongation complex (TEC) into this framework, but it does so with less depth. The observation that sRNA can bind to the TTC-LC complex is interesting and may inform future studies of sRNA regulation. However, the scope is narrow, and important mechanistic questions remain unaddressed. For example, whether LC size influences sRNA binding, or whether the binding kinetics are compatible with the timescale of TTC formation, is not investigated. Ribosome transit through the mRNA could strongly affect sRNA binding, yet no measurements were provided to assess this.

- Nascent RNA degradation in the early expressome. Given that E. coli RNase E is membrane-associated, it is not clear how physiologically relevant it is to study RNase E activity on nascent mRNAs in this in vitro setting. The inclusion of RNase E here feels somewhat forced. Moreover, in Figure 4C-D, the primary determinant of degradation kinetics appears to be the presence or absence of the 30S IC, whereas the contribution of the TEC is comparatively minor.

We thank the reviewers for careful reading of the manuscript and the helpful comments. We have replied to the points below.

Reviewer #1 (Remarks to the Author):

The current view is that transcription and translation are coupled in many prokaryotic species. Furthermore, the cooperation between RNA polymerase (RNAP) and the ribosome may have functional consequences that are difficult to predict. However, the steps involved to establish coupling between RNAP and the pioneering ribosome are still poorly understood. This manuscript by Roske, Paris, et al. describes cryo-EM reconstructions of *E. coli* small ribosomal subunit initiation complexes (30S IC) and a paused RNAP and explores how the *ompD* transcript is regulated via degradation by a small regulatory RNA (sRNA) called MicC. The authors describe reconstructions of two 30S ICs in isolation, an RNAP transcription elongation complex (TEC) assembled on the putative *ompD* pause site, as well as complexes of the TEC coupled to the 30S IC via the transcription factor NusG (and potentially also NusA). Overall the manuscript is well written but I have several concerns and some aspects would benefit from clarifications and additional experimental evidence (see below) before publication can be considered.

Major comments:

An important concern is that the authors decided to assemble a TEC at two different positions with respect to the MicC binding site to test RNA degradation (OmpD99 and OmpD105). If I understand correctly, the TEC at *ompD*105 was then also used for the complex between 30S-IC and the TEC. All of this relies on the fact that the *ompD*105 TEC recapitulates the paused RNAP but this is where I am skeptical.

1. The authors decided to alter the RNA/DNA hybrid sequence compared to the wildtype situation (see their supplementary Figure 2F, and 2G). As pointed out by the authors themselves, the RNA 3'-end, and the next template and/or non-template DNA base are believed to be critical to induce pausing in a TEC (see their supplementary Figure 2A). This reviewer is puzzled why the authors decided against using the natural sequence, which supposedly induces a paused TEC conformation. The concern is of course that the TEC is not adopting a paused conformation and thus skewing the results.

2. Further, the *ompD*-TEC sequence as depicted in Figure 2D does not match the one depicted in supplementary Figure 2F and neither of them matches the wildtype sequence of the putative pause site as depicted in supplementary Figure 2B. Please explain.

Our reply: Although the putative pause site is functional (see Supplementary Figure 2C), paused complexes from *in vitro* transcription using RNAP did not result in a stable specimen suitable for cryo-EM or degradation assays in our hands. We therefore chose to reconstitute TECs on a pre-formed transcription bubble scaffold containing non-complementary segment in the DNA substrate that allows almost complete engagement of RNAP in the TEC. This, however, inevitably leads to changes of the properties of a TEC, since not only the sequence of the template DNA-mRNA hybrid, but also the single-stranded segment of the non-template strand is suggested to play important roles in transcription stalling, e.g. through thermodynamics and direct recognition by transcription elongation factors.

We note that prolonged RNA polymerase pausing is not a critical aspect of our proposed model for mRNA exposure to sRNA, all that is required is a loop to form; and the recent results of Duss (2025) indicate that this can occur in transcription-translation coupling (Qureshi and Duss, 2025). Whilst we identified a putative pause site around position +106 in *ompD*, we also tested several other positions of paused TECs around the MicC recognition site in mRNA degradation reactions (data included in the updated manuscript). To ensure that the obtained results in these reactions are dependent on the TEC's position rather than the efficiency of TEC formation, we decided to use the same DNA scaffold and 3' mRNA segment at every position. The scaffold sequence is an adaptation from previous reports of TEC reconstitutions (Kohler et al., 2017, Webster et al., 2020, Said et al., 2021) to optimise efficiency and homogeneity of TEC formation, whilst avoiding complementarity with upstream regions in the *ompD* mRNA constructs. Our cryo-EM experiment shows that the chosen sequence stabilises the TEC in the post-translocated state, improving homogeneity for structural studies.

We thank the reviewer for noting the discrepancies between the sequences in the mentioned Figures. During review of our structural data, we were able to assign the exact sequence register of the reconstituted TECs based on the distinction of pyrimidine and purine bases and have amended the schematic Figures accordingly.

Lastly, we have changed the description in Supplementary Fig. 2 to be clearer.

3. The authors make little comment about the *ompD*105-TEC reconstruction. Does it exhibit any of the characteristics that have been observed in paused TECs (swivelling, RNA/DNA hybrid tilting, more pre- as opposed to post-translocated register)?

Our reply: Overall, our TEC reconstruction shows very little conformational variability. All classes display a post-translocated register. We have included a video clip of the 3D variability analysis to show conformational changes across all high-resolution particles (**Movie M1**). Since this is outside the scope of the study, especially because the TEC was reconstituted on an artificial nucleic acid scaffold and does therefore not represent the nature of a TEC positioned at the putative pausing site (see replies above), we omitted a detailed description in the manuscript.

4. Why would the *ompD*105-TEC lack omega but not the *ompD*105-TEC-30S-IC – is there any difference in preparation that could explain this discrepancy?

Our reply: The absence of the omega subunit in the TEC structure shown here might be due to loss of the subunit during complex purification via gel filtration and/or during the grid freezing process on the graphene oxide support surface. We have solved the *ompD*-TEC structure again from the same RNAP preparation but omitting gel filtration and graphene oxide surface coating of the grids, and instead using CHAPSO detergent during sample vitrification to overcome preferred orientation (data included in the updated manuscript). The reconstruction shows the *ompD*-TEC in the same conformation and with very good occupancy of the omega subunit. This supports the hypothesis that the lack of density for the omega subunit is due to differences in preparation of cryo-EM specimen. We think it unlikely that the dynamics of the omega subunit are influenced, i.e. stabilised, by the presence of 30S-IC.

5. There is very little to no comment on NusA in the reconstructions and biochemistry. Does NusA contact the 30S-IC? If so, is it in a similar fashion to what has been reported for expressomes (see Wang et al., Science 2020).

Our reply: We observe NusA NTD contacting only the RNA polymerase-portion in the coupled TEC-30S IC complex in a similar manner to previously reported TEC structures containing NusA (Guo et al., 2018). The resolution is very low, and the details cannot be noted with confidence, but additional unoccupied density in the TEC structure that arose from the 30S IC-containing dataset can contain S1 motif and possibly KH domains that ensue the NusA NTD. We do not observe density for NusA on the surface of the 30S IC. We have added a label to Figure 3A to note the position of NusA NTD and expanded the description of NusA in the revised text as above.

6. The authors suggest that the 30S and TEC facilitate recognition of *ompD* transcript for degradation. My first thought was, since degradation rates are reduced in presence of 30S and TEC (Figure 4B and 4C) that the gene expression machinery provides protection – wouldn't that make more sense? As long as the mRNA is transcribed (and more importantly) translated, it is protected from degradation? This agrees with the reduced rate of overall *ompD* degradation and also reduced rate of +83 intermediate degradation, no? Also, it is unclear to this reviewer how specificity for +83 is measured (Figure 4D)?

Our reply: We appreciate the point made by the reviewer about protection, and we have changed the text to explain this better. The naked RNA used in *in vitro* experiments is a control reaction and is not expected to reflect the *in vivo* state, where RNA is always bound by proteins, therefore it is expected that gene expression machinery provides overall protection.

MicC-dependent regulation of *ompD* has been shown to initiate with cleavage at position +83 (Pfeiffer et al., 2009), which serves as the key mRNA inactivation event we focused on. Therefore, despite the overall protection of the mRNA by bound proteins, the sRNA must have an opportunity to access its target. Here, we demonstrate that the +83 site is accessible in TEC, 30S IC and coupled assemblies, and that cleavage preference for the +83 position is increased in the complexes, suggesting that the 30S and TEC might help with presentation of the target site, while protecting *ompD* from off-target and unspecific cleavage and degradation. With our updated manuscript, we include additional experiments; degradation reactions done on reconstituted TEC complexes at different positions along *ompD*. These experiments show that the increased efficiency of RNase E activity on the MicC-induced site is dependent on the position of the RNAP.

The calculation of +83 cleavage specificity determined as described in the legend of Figure 4 and the text (line 333-335): '[...]cleavage specificity for the +83 site [...] is expressed as the formation rate of the +83 intermediate over the degradation rate of the *ompD* starting material. Cleavage specificities for the +83 site in the different *ompD* assemblies are shown in Figure 4D.' We have added reference to Fig. 4C as a source of rates additional clarification.

Minor comments:

The authors mention that the 30S head exhibits less movement in absence of IF3 in

the 30S-IC reconstructions – what is the metric for this observation? Did the authors perform 3D variability analysis and compared the extreme head positions? Please clarify in the manuscript.

Our reply: We have compared the movement of the 30S head by 3D variability analysis. We have clarified this point in the revised manuscript (Results segment 1 and Materials and Methods) and have included a Supplementary Movie to illustrate the mobility of the 30S head in the two structures with the revised submission.

The authors mention that their bS1 conformation (I suggest to use the revised nomenclature for all ribosomal proteins – i.e. bS1 not S1, uS10 not S10, etc. – see Ban et al., Current Opinion in Structural Biology, 2014, PMID 24524803) is different from expressome structures but similar to a recent report of complexes between RNAP and the 30S subunit (Webster et al., Science 2024, PMID 39607923). What exactly is different? As far as I know, bS1 has either not been modelled in the available expressome structures or only the N-terminal helix and the first OB domain of bS1 have been deposited. I believe this reflects the fact that bS1 is mobile and only OB1 is in a somewhat stable orientation with respect to the rest of the 30S and is consistent with Webster et al., Science 2024.

In that sense, the statement of the authors, which suggests a difference in bS1 conformations between 30S and 70S ribosomes/expressomes is misleading and should be removed or further clarified (what part of bS1 is different and compared to which reconstruction exactly).

Our reply: We have revised the nomenclature for the ribosomal proteins throughout the revised manuscript.

In our structures, bS1 is largely mobile, except for the two N-terminal OB fold-like S1 domains. We have further clarified the text here.

Figure 1 shows cryo-EM reconstructions of two 30S populations (30S-IC with IF1 and IF3, and 30S-IC with IF1, IF2, and fMet-tRNA^{fMet} – it is fMet-tRNA^{fMet}, i.e. aminoacylated and formylated initiator tRNA, correct?). Are these consensus reconstructions? If so, the authors may consider to perform focused refinement of 30S head and body to obtain higher local resolution.

Our reply: The initiator is aminoacylated and formylated, and we have replaced wording with fMet-tRNA^{fMet} throughout the manuscript. The reconstructions are consensus refinements. Focused refinements did indeed result in improved local resolution for the 30S head, but did not significantly improve the regions for initiation factors, tRNA, mRNA, or bS1, which is why we did not include these maps in the main submission. We have included more detail on Cryo-EM data processing and image sorting in the Material and Methods section of the revised manuscript.

As mentioned before, the ompD-TEC cryo-EM reconstruction (Fig. 2A-C) lacks the RNAP omega subunit according to the authors. I noticed that E. coli RNAP was expressed from a plasmid that encodes all 5 RNAP subunits but no additional plasmid was used to express the omega subunit. I seem to remember anecdotal evidence that rpoZ needs to be overexpressed from an additional plasmid to ensure

stoichiometric amounts and was curious if the authors have considered this to explain their observation.

Our reply: We thank the reviewer for this suggestion and will consider this approach in our future experiments. As we describe above, we observe clear cryo-EM density for the omega subunit in a TEC reconstitution from the same RNAP stock. In this reconstitution, the gel filtration step is omitted, when EM specimen is prepared at a much higher sample concentration (8-10 μ M) vitrified in the presence of CHAPSO detergent, rather than on graphene oxide-coated grids. We therefore assume that the omega subunit is present in near stoichiometric amounts in our RNAP sample but might not be visible in the TEC structure due to differences in TEC reconstitution and/or EM specimen preparation.

Supplementary Figure 1C: the legend still contains a comment from editing and the map is segmented but not coloured by local resolution as stated in the legend.

Our reply: We have corrected these in the revised text.

Supplementary Figure 1D: authors should include particle orientation plot

Our reply: The plot has now been incorporated as Suppl. Figure 6.

Supplementary Figure 1E: the map is segmented but not coloured by local resolution as stated in the legend.

Our reply: This has been corrected.

Supplementary Figure 1F: authors should include particle orientation plot

Our reply: The plot has now been incorporated as part of Suppl. Figure 6.

Supplementary Figure 2C-2D: The pause signal is very weak and the authors may miss the time-point when the paused species reaches its maximum. A time course would be much more useful to appreciate accumulation of paused species and decay as RNAP escapes from the pause. There is also no obvious effect of NusA and/or of temperature on the extent of pausing – again, a time course (i.e. RNA transcription kinetics) would reveal an effect. In S2D it is unclear which lane was used to generate the profile – is this an average of all 4 lanes? That's not very useful in my opinion. In general, a radioactive (32 P) assay may be superior to staining with SYBR gold. It is also unclear to me why the authors decided to flank the ompD sequence with vector-derived sequence? Why not use the regular ompD sequence – this would avoid any potential effect of upstream sequence on pausing and would also make analysis easier because the paused species is shorter, easier to resolve, and its length could be more reliably measured on high-resolution denaturing PAGE. This comment also reflects another concern I highlighted earlier: the fact the authors use inconsistent sequences is a problem in my opinion: i) the sequence for the assays is different from the sequence used for the cryo-EM reconstructions; and ii) the terminal 9 nucleotides used for cryo-EM reconstructions does not reflect the ompD pause sequence.

Our reply: We acknowledge the reviewer's concern that the transcription reaction is a multi-round one, and the picture shows the steady-state amount of the pause. The kinetics of transcription pausing would be interesting for detailed study of *ompD*, but we feel they are beyond the scope of our study, and we do not believe that they would strengthen the presented data. We have added clarification to the legend of the referenced Figure that the top panel shows averaged signal of all four *in vitro* transcription reactions. Regarding the sequences for reconstitution of stable TEC complexes, please see the reply above. In our proposed model pausing is not critical for general sRNA access to exposed loops of mRNA, which becomes possible once the recognition site on the mRNA exits the RNAP. We have changed the text throughout the manuscript to place less emphasis on pausing.

Supplementary Figure 2M,N: The legend says *ompD*105 but the figure says *ompD*99. The bottom plot should presumably say *ompD*83 not *ompD*99? Also, the authors propose that the TEC protects *ompD*99 from cleavage mediated by MicC yet *ompD*99 degradation is not affected according to the plot in S2N – please elaborate. Further, I think it would be fair to include *ompD* RNA alone as a negative control so we can appreciate that the RNA is stable in absence of RNase E under these reaction conditions. Is the RNA quantified as fraction of total signal in each lane?

Our reply:

We thank the author for the discrepancies between figure legend and plot labels. We have amended these in the updated manuscript.

Regarding cleavage at position +83, the *ompD*99-TEC becomes protected by RNAP because the cleavage site is made inaccessible by the RNAP. Although the degree to which the +83-cleavage intermediate is accumulated becomes reduced by almost 3-fold, this difference reflects only ~3% of the starting material and the measurement of the degradation rate of the starting material is not sensitive enough to reflect these changes.

The RNA signal was quantified by *ompD* signal in each lane and normalised to time zero lane. The RNA only control is lane t = 0 which was taken before addition of RNase. We included detailed description of the degradation assay procedure in Material and Methods. RNAs, after removing from storage at -20 °C are routinely heated to 50 °C and slowly cooled to 37 °C for biochemical assays. For degradation reactions, mRNA, sRNA and Hfq are pre-incubated together in reaction buffer and at 37 °C for 15 min prior to addition of RNase. Time points 0 min in presented gels represent aliquots of the pre-incubations, which confirms that they are stable in the absence of RNase E.

I would encourage the authors use the number of nucleotides rather than codons to describe the distance between RNAP and the ribosomal P-site because RNAP moves in steps of single nucleotides and there may not be an integral number of codons them.

Our reply: Codons were used in previous studies of Expressomes, measuring distances between RNAP mRNA exit channel and ribosome. For this reason we maintain the use of codons because we are measuring in the open reading frame.

Reviewer #2 (Remarks to the Author):

The authors report a cryo-EM structure of an E. coli transcription-translation complex (TTC) comprising a promoter-proximally paused E. coli transcription elongation complex (TEC) and a ribosome 30S subunit translation initiation complex, coupled by mutual interaction with the transcription elongation factors NusA and NusG.

The authors further show that, in this complex, the mRNA spacer between the TEC and the ribosome 30S subunit is accessible to interactions with a small regulatory RNA and to cleavage by the RNA degradosome assembly.

The authors infer that physical coupling between transcription and translation can be established at the initiation stage of translation and that the mRNA spacers in coupled complexes at the initiation stage of translation are accessible to regulatory factors.

The structural organization of the TEC and ribosome in the authors' complex matches, in detail, the structural organization of the recently reported long-range transcription-translation complex (TTC-LC; <https://www.biorxiv.org/content/10.1101/2024.07.20.604413v1>), and the inferred functional role of the authors' complex and accessibility to regulators of the authors' complex matches the functional role and regulatory accessibility recently proposed for TTC-LC (<https://www.biorxiv.org/content/10.1101/2024.07.20.604413v1>).

The results are important and timely and warrant publication. However, prior to publication, the manuscript must be revised to cite and describe the recent structural work on TTC-LC and to cite and describe the proposed functional role and regulatory accessibility of TTC-LC (<https://www.biorxiv.org/content/10.1101/2024.07.20.604413v1>). The recent structural work on, and proposal about, TTC-LC provide crucial context that makes the authors' work important and timely, but, surprisingly, the current manuscript does not cite or describe the previous work.)

Our reply: We note that all expressome structures are not static but have multiple conformations. This is especially the case for expressome structures with long mRNA linkers between RNAP and ribosome. The TTC-LC study referenced by the reviewer and our structure now show that the physical link between RNAP and 70S or 30S (respectively) can be established despite such long mRNA linkers. We have revised the manuscript to cite the recent work on the TTC-LC.

Specific comments

28: Add "and transcription factor NusA" (unless evidence is obtained and presented that NusA does not contribute to stability of the complex).

Our reply: We have added the text.

57: Add citations

to <https://pubmed.ncbi.nlm.nih.gov/39117885/> and <https://www.biorxiv.org/content/10.1101/2024.07.20.604413v1>. Delete citation to Qureshi & Duss, 2025 (which does not report a cryo-EM structure).

Our reply: We have added the citations, but kept the citation to Qureshi and Duss, because the sentence has been changed to include both structural and single molecule results (see next point raised by the reviewer).

61: Replace "structural" by "structural and single-molecule

Our reply: We have changed the text as suggested.

65-66: Add citations

to <https://pubmed.ncbi.nlm.nih.gov/39117885/> and <https://www.biorxiv.org/content/10.1101/2024.07.20.604413v1>.

Our reply: We have added the citations to the revised text.

83: Cite and describe hypothesis that translation initiation and ribosome catch-up involve a long-range, loosely coupled transcription-translation complex (TTC-LC, with a long, >11 codons, looped-out mRNA spacer between TEC and ribosome, and with accessibility of the mRNA spacer to regulatory factors), and that subsequent coupled transcription and translation involves a short-range, tightly coupled transcription-translation complex (TTC-B, with a different orientation of ribosome relative to TEC, with short, 7-11 codons, mRNA spacer, and with little or no accessibility of the mRNA spacer to regulatory factors)(<https://www.biorxiv.org/content/10.1101/2024.07.20.604413v1>).

Our reply: We have added the citation and added the text, with the caveat that the comparison is between the 70S particle in the TTC-LC, while in our study we are looking at the translation initiation complex without the 50S.

92: State that the complex obtained has the same orientation of ribosome S30 subunit to TEC as in TTC-LC (<https://www.biorxiv.org/content/10.1101/2024.07.20.604413v1>).

Our reply: We have added the citation and added the text.

93: Add, "as proposed for TTC-LC

in <https://www.biorxiv.org/content/10.1101/2024.07.20.604413v1>."

Our reply: We have added the citation and added the text.

280-286: Is cryo-EM density for the NusG interdomain linker present? If not, the linker should be omitted--or should be rendered or colored differently--in Fig 3B.

Our reply: The density is not present, and we have changed Figure 3B (now 3A) to make the linker a dashed line.

286: Add text describing density and interactions for NusA (which is shown to be in the complex in Fig 3B, but is not discussed in the current text).

Our reply: The resolution is not sufficient to describe the interactions, but we note that the density fits well to NusA and is in the same relative position as seen in earlier studies, to which we refer. We have expanded the discussion of the NusA in the revised text, as we do not observe NusA on the surface of the 30S IC.

289-292: Perform and report results of analogous analysis assessing whether formation of the complex is dependent on NusA.

Our reply: In the experiments described in line 289-292, we did add both NusA and NusG. We have changed the text to make this more clear. We did not explore if NusA is a key player in the mechanism of sRNA binding site exposure. The role of NusA is interesting, but it is not the focus of the present study. We refer in the revised text earlier studies that report on the requirement of NusA for expressosome formation.
(Webster et al. 2024).

297-299: Add figure panels to Fig 3 showing superimpositions of the authors' complex on TTC-B and TTC-LC. (Panels A, C, and D of Fig 3 can be moved to the Supplement, if additional space is needed.)

Our reply: We note that the TEC and 30S-IC are highly flexible with respect to each other, and for this reason it may not be so informative to show their superpositions. The conformation that we show in the figure is arbitrarily chosen for the composite map. The 2D classes in panel C and D are in the main Figure 3 to illustrate this.

297-299: Replace current text--which is incorrect unless "general" in "general orientation" is interpreted *very* loosely--with text stating that the the orientation of ribosome 30S subunit to TEC is different from that in TTC-B but is the same as that in TTC-LC.

Our reply: We have replaced the text.

300: Add paragraph on the mRNA spacer between ribosome S30 subunit and TEC. For what part or parts of the mRNA spacer is cryo-EM density detected? What inferences about the path of the spacer mRNA can be made? For example, can it be inferred that the mRNA spacer is too long to follow a straight path and therefore must be looped out?

Our reply: The map is not sufficiently clear to trace the nucleotides of the >70nts spacer, but the model and map indicate that the mRNA spacer is too long and must be looped out.

344-347: Add "and transcription factor NusA" (unless evidence is obtained and presented that NusA does not contribute to stability of the complex).

Our reply: We have not done experiments to address if NusA is required. This will be the subject of another study. We changed the text accordingly.

387-389: State that the complex obtained has the same orientation of ribosome S30 subunit to TEC as in TTC-LC (<https://www.biorxiv.org/content/10.1101/2024.07.20.604413v1>).

Our reply: The revised text now includes a statement on the relative orientation between TEC and 30S, which is highly flexible (as indicated by multi body refinement + principal component analysis of the relative orientations between bodies, shown in Figure 3C and Suppl. Movie 2).

391: Delete "similar." Clarify the differences.

Our reply: We have deleted the word similar and commented on the differences.

Reviewer #3 (Remarks to the Author):

In this work, cryo-EM is used to investigate complexes formed on *ompD* mRNA, which is subject to regulation by the small RNA, MicC, in Salmonella. Structures of 30S initiation complexes (one with IF1 and IF3; another with IF1, IF2 and fMet-tRNA) were determined, as were structures of the 30S IC and RNAP. For the latter, RNAP was positioned via a heteroduplex (bubble) template ~100 base pairs downstream from the start codon. RNAP and the 30S subunit interacted with one another through NusG / NusE (S10), with mRNA (~70 nt) presumably looped out (unresolved) between the two machines. The MicC recognition site is positioned in the middle of this looped-out mRNA. Treatment of these complexes with RNase E and MicC/Hfq resulted in mRNA decay, the kinetics of which differed depending on which complex was present. While this work provides some impressive structural data, including for example new views of S1, the biological relevance here is unclear and the paper is dominated by speculative text.

Specific concerns:

1. Pfeiffer et al 2009 showed that transplantation of the MicC binding site into another gene (*ompN-gfp*) leads to MicC-dependent control, akin to that seen for the native gene. This argues against a complicated mechanism. Is there any evidence that MicC-dependent control of *ompD* requires transcription-translation coupling? Without such evidence, the rationale for this work is unclear.

Our reply:

The *in vivo* effects may be confounded by many factors, making it challenging to definitively test and dissect the underlying mechanisms. For instance, the regulation of *ompN-gfp* in the presence of overexpressed sRNA may bypass the natural regulatory mechanisms that typically govern MicC activity under normal conditions. However, we are not aware of any evidence for *ompD* strictly requiring transcription-translation coupling, and we are not suggesting that coupling is obligatory for MicC action. Instead, we suggest that the coupling does not prevent regulation and may offer an opportunity for sRNA action, and this possibility is supported in principle by *in vitro* observations. We also note that the access can analogously occur at the 5'

leading edge of the polyribosome. Our complex and proposed accessibility to regulators to the complex are in accord with the functional role and regulatory accessibility recently proposed for TTC-LC by the Ebright group (Wang et al., 2025 biorxiv).

2. In the RNase E assay, mRNA decay is seen whether it is MicC-guided or not. This makes interpretation of the data tricky, to say the least. The authors state that the “+72 cleavage product is an off-target species only observed *in vitro*.” How then do we (the readers) interpret overall decay rates in these experiments? In the cell, there is a large difference in decay rates (+/- MicC), which this *in vitro* assay fails to recapitulate.

Our reply: The key difference between *in vitro* assays and the *in vivo* environment is the presence of numerous cellular factors that interact with the RNA. *In vitro*, we examine isolated components of the system, leaving certain regions of the RNA exposed and susceptible to RNase E cleavage. This explains why the degradation of *ompD* does not depend on the presence of MicC. However, the +83-cleavage product, which has been shown to initiate *ompD* degradation *in vivo* (Pfeiffer et al., 2009), is observed *in vitro* only when MicC is present. In our assays we interpret the decay rates only in the context of accumulation of MicC-dependent product (+83 cleavage product), i.e. as an indicator of specificity for the MicC-guided cleavage site.

3. The authors speculate a lot about transcription-translation coupling and its potential role in targeted mRNA decay. Yet, no experiments to disrupt coupling and monitor the consequences in the cell are presented.

Our reply: We have reduced the speculation and focus more on the looping and its potential accessibility. As noted above, the results are consistent with looping observed by Wang et al., biorxiv. Moreover, we do not propose that coupling is essential for MicC-guided RNA regulation. Instead, we suggest that regulation can still occur despite the formation of TIC or TEC, meaning that the sRNA can access its target gene even during the coupling. We have changed the text to make this clearer.

4. In the RNAP-NusG-30S complexes, the authors envision that a single mRNA is bound. Has this been verified? Can similar NusG-linked complexes be made with two separate halves of this mRNA? If not, why not? Addressing these questions could increase the impact of this work substantially.

Our reply: While this is a theoretical possibility, we have not explored if the NusG complex can be made by two separate halves of the mRNA. For coupled complexes, 30S-IC were reconstituted on pre-assembled TEC. For TECs, RNAP was added in 1:1 ratio to RNA. We note that mRNA-free 30S or RNAP particles are rare in the images.

5. Unclear is whether the pause site in *ompD* was uncovered by genome-wide experimental data or predicted by the consensus motif derived from such data.

Our reply: The putative pause site was identified by *in vitro* transcription and match to consensus motif, and we have changed the text to make this clearer.

6. Figure 5 is complicated and seems more appropriate for a review on RNase E than for this study. Also, the localization of RNase E in *E. coli* seems at odds with case (i), the case studied here (what?!?). The extended portion of RNase E (purple string) is difficult to distinguish from the mRNA (blue string).

Our reply: We have simplified Figure 5. Just to explain further, case (i) would be more likely to occur in a bacterium such as *Caulobacter crescentus*, where RNase E is not localised to the cytoplasmic membrane. We also note that the access, as proposed for the coupled transcription-translation intermediate, can analogously occur at the 5' leading edge of the polyribosome and the translation initiation complex.

7. In the Intro, the authors state: "Transcription-translation coupling (TTC) appears to occur extensively in representative gram-negative bacterial species,..." However, an analysis of six operons in *E. coli* showed evidence for transcription-translation coupling in only one operon and only in the presence of a mutation that slowed down RNAP, suggesting that coupling is uncommon and stochastic (PMID 30275301). Moreover, Hwa and co-workers provided compelling evidence that coordination of transcription and translation in *E. coli* is mediated indirectly via the concentration of ppGpp and aminoacyl-tRNA (PMID 31451774). Yes, several structural studies have shown how NusG can link RNAP and the ribosome. But, how frequently RNAP and the ribosome are coupled in *E. coli* and what role this physical interaction plays remain open questions.

Our reply: We thank the reviewer for raising this point, and we have added the citations to the revised text and made the speculation text more cautious. The study by Zhu et al (2019) (PMID 31451774) provides evidence that translation is not required to maintain the speed of transcriptional elongation, but the results of single molecule studies by Qureshi & Duss, 2025 indicate that long loops can form that can maintain the coupling. We note that the coupling discussed by Chen and Fredrick 2018 (PMID 30275301) refers to the 70S, but the study here is for the 30S preinitiation complex.

- GUO, X., MYASNIKOV, A. G., CHEN, J., CRUCIFIX, C., PAPAI, G., TAKACS, M., SCHULTZ, P. & WEIXLBAUMER, A. 2018. Structural Basis for NusA Stabilized Transcriptional Pausing. *Mol Cell*, 69, 816-827 e4.
- KOHLER, R., MOONEY, R. A., MILLS, D. J., LANDICK, R. & CRAMER, P. 2017. Architecture of a transcribing-translating expressome. *Science*, 356, 194-197.
- QURESHI, N. S. & DUSS, O. 2025. Tracking transcription-translation coupling in real time. *Nature*, 637, 487-495.
- SAID, N., HILAL, T., SUNDAY, N. D., KHATRI, A., BURGER, J., MIELKE, T., BELOGUROV, G. A., LOLL, B., SEN, R., ARTSIMOVITCH, I. & WAHL, M. C.

2021. Steps toward translocation-independent RNA polymerase inactivation by terminator ATPase rho. *Science*, 371.

WEBSTER, M. W., TAKACS, M., ZHU, C., VIDMAR, V., EDULJEE, A., ABDELKAREEM, M. & WEIXLBAUMER, A. 2020. Structural basis of transcription-translation coupling and collision in bacteria. *Science*, 369, 1355-1359.

Reviewer #1 (Remarks to the Author):

I have been asked to comment on the author's response to comments from both reviewer 1 as well as reviewer 3. Reviewer 3 seemed particularly concerned about the overall impact and relevance.

1.: The authors need to clearly specify that the TEC has not been assembled on the natural pause-inducing sequence but instead on a scaffold with a different sequence to stabilize the TEC. The reader needs to be aware of this potential limitation. It is debatable if a TEC on a non-pause-inducing sequence and a paused TEC are the same and I would argue readers should decide themselves - thus claiming that the TEC mimics a paused complex is not accurate in my opinion and should be avoided. However, I understand perfectly well that the state of the TEC (paused or not) may not have any influence on RNA cleavage. This is really a question of using accurate wording to be clear avoid misunderstandings.

Our reply: We have changed the wording in the text to make it clear that the TEC is assembled on a different scaffold for stability.

2.: Minor comment: I believe the vast majority of studies in the translation, transcription, and coupled translation-transcription field use nucleotides to denote distance along a (shared) mRNA. Recent work by the Duss lab, Weixlbaumer lab, or Zenkin lab, which addressed transcription-translation coupling also used that metric. However, I am sure readers can multiply the number of codons by 3 if the authors prefer to use codons as their preferred unit.

Our reply: We have used nucleotides to denote distance.

3.: It remains puzzling why the authors decided to use a coupled transcription-translation system to study targeting by MicC because it seems MicC can target the RNA regardless of whether it is free, still bound to a TEC, or bound to a TEC and a 30S-IC? It would thus be useful to emphasize the rationale of the study more as pointed out by reviewer 3.

Our reply: MicC targets the coding region of *ompD* mRNA, and it has been unclear at which stage of gene expression this sRNA can effectively act. The aim of the study was to determine whether sRNAs can, in principle, interfere with gene expression even at late stages of translation initiation to suppress protein synthesis. We have added this clarification to the revised text.

Additional comments:

The abstract says "...RNA polymerase that is paused proximally to the promoter can associate with the pioneering 30S translation initiation complex (30S IC) through mutual binding of the transcription factors NusG and NusA." However, the authors were unable to resolve NusA bound to the 30S.

Our reply: We have changed the wording of the abstract.

Reviewer #2 (Remarks to the Author):

The revised manuscript fails to address key points raised in my previous review:

92: State that the complex obtained has the same orientation of ribosome S30 subunit to TEC as in TTC-LC

(<https://www.biorxiv.org/content/10.1101/2024.07.20.604413v1>).

Our reply: The model is not available and we cannot state this confidently. We have however changed the text to comment on the similar orientation..

93: Add, "as proposed for TTC-LC

in <https://www.biorxiv.org/content/10.1101/2024.07.20.604413v1>."

Our reply: This is now stated in the revised manuscript.

289-292: Perform and report results of analogous analysis assessing whether formation of the complex is dependent on NusA.

Our reply: This is beyond the remit of the study, and we have changed the wording here to make it clear that NusA can be involved, but has not been tested.

297-299: Add figure panels to Fig 3 showing superimpositions of the authors' complex on TTC-B and TTC-LC.

Our reply: We appreciate the suggestion, however, we believe that including this information would not substantially enhance the report. The models have been deposited and can be readily compared with other complexes in the database.

300: Add paragraph on the mRNA spacer between ribosome S30 subunit and TEC. For what part or parts of the mRNA spacer is cryo-EM density detected? What inferences about the path of the spacer mRNA can be made? For example, can it be inferred that the mRNA spacer is too long to follow a straight path and therefore must be looped out?

Our reply: Unfortunately, the map resolution is not sufficient to model the linker with confidence, which limits our ability to expand this part of the discussion in a meaningful way.

387-389: State that the complex obtained has the same orientation of ribosome S30 subunit to TEC as in TTC-LC

(<https://www.biorxiv.org/content/10.1101/2024.07.20.604413v1>).

Our reply: This is now stated.

Prior to possible publication, these key points will need to be addressed.

In addition, the current Abstract (which fails to state the results and conclusions of the work) will need to be revised, and the current text at 527-528 (which misrepresents a hypothesis as a fact and fails to state the rationale for the hypothesis) will need to be revised.

Our reply: We have revised the abstract and text.

Reviewer #4 (Remarks to the Author):

This manuscript addresses important questions in bacterial RNA biology, including transcription-translation coupling, sRNA regulation, and RNA degradation. They used cryo-EM approaches alongside biochemical assays. Below, I summarize the main contributions of the study as well as my reservations.

- Structural evidence of an early expressome, composed of RNAP and 30S IC (+/- NusG). While this finding is of interest, it is not entirely novel. A similar complex was previously resolved and reported by Webster et al. (Science, 2024). The structural details are beyond my expertise, but the degree of novelty relative to that prior work appears limited.

Our reply: We are aware that Webster et al. (2024) reported similar structures but in a pre-active state, proposing two pathways for initiating transcription-translation coupling through transcription-assisted recruitment of mRNA to the ribosome. However, our data include initiation factors IF1, IF2, and IF3 and capture the 30S initiation complex in fully 'accommodated' active state, which could arise through either of the pathways proposed by Webster et al.

- sRNA binding within the early expressome. This extends the authors' prior work (Bandyra et al., NAR 2024), which already provided a thorough examination of RNA degradation kinetics in the presence of MicC and the 30S IC in vitro. The present study adds the transcription elongation complex (TEC) into this framework, but it does so with less depth. The observation that sRNA can bind to the TTC-LC complex is interesting and may inform future studies of sRNA regulation. However, the scope is narrow, and important mechanistic questions remain unaddressed. For example, whether LC size influences sRNA binding, or whether the binding kinetics are compatible with the timescale of TTC formation, is not investigated. Ribosome transit through the mRNA could strongly affect sRNA binding, yet no measurements were provided to assess this.

Our reply: We now mention these limitations in the revised manuscript.

- Nascent RNA degradation in the early expressome. Given that E. coli RNase E is membrane-associated, it is not clear how physiologically relevant it is to study RNase E activity on nascent mRNAs in this in vitro setting. The inclusion of RNase E here feels somewhat forced. Moreover, in Figure 4C-D, the primary determinant of degradation kinetics appears to be the presence or absence of the 30S IC, whereas the contribution of the TEC is comparatively minor.

Our reply: We appreciate the reviewer's comment and agree that the cellular context of RNase E localization can influence its accessibility to substrates. However, RNase

E is not membrane-associated in all bacteria, such as *Caulobacter crescentus*, and therefore the proposed model remains relevant in these cases. Moreover, even in *E. coli*, the initiation complex can occupy the leading position within a polysome and thus remain accessible to the membrane-associated enzyme. With respect to the kinetic data, our main conclusion concerns not only the overall rate of *ompD* degradation but rather the efficiency of the mRNA-inactivating cleavage at position +83, which occurs in vivo in *Salmonella* Typhimurium upon MicC expression. Considering this, TEC alone enhances cleavage efficiency, whereas the presence of the 30S IC appears to reduce it.